# Pharmacological inhibition of HDAC6 improves muscle phenotypes in dystrophin-deficient mice by downregulating TGF-β via Smad3 acetylation

Alexis Osseni [1,2] ✉, Aymeric Ravel-Chapuis [3], Edwige Belotti[1], Isabella Scionti[1], Yann-Gaël Gangloff [1], Vincent Moncollin[1], Laetitia Mazelin[1], Remi Mounier[1], Pascal Leblanc[1], Bernard J. Jasmin [3,4,5] ✉ & Laurent Schaeffer [1,2,5] ✉

The absence of dystrophin in Duchenne muscular dystrophy disrupts the dystrophin-associated glycoprotein complex resulting in skeletal muscle fiber fragility and atrophy, associated with fibrosis as well as microtubule and neuromuscular junction disorganization. The specific, non-conventional cytoplasmic histone deacetylase 6 (HDAC6) was recently shown to regulate acetylcholine receptor distribution and muscle atrophy. Here, we report that administration of the HDAC6 selective inhibitor tubastatin A to the Duchenne muscular dystrophy, *mdx* mouse model increases muscle strength, improves microtubule, neuromuscular junction, and dystrophin-associated glycoprotein complex organization, and reduces muscle atrophy and fibrosis. Interestingly, we found that the beneficial effects of HDAC6 inhibition involve the downregulation of transforming growth factor beta signaling. By increasing Smad3 acetylation in the cytoplasm, HDAC6 inhibition reduces Smad2/3 phosphorylation, nuclear translocation, and transcriptional activity. These findings provide in vivo evidence that Smad3 is a new target of HDAC6 and implicate HDAC6 as a potential therapeutic target in Duchenne muscular dystrophy.

Duchenne muscular dystrophy (DMD) is an X-linked neuromuscular recessive disorder affecting approximately 1 in 3500 newborn males worldwide and is the most common and fatal form of muscular dystrophy[1,2]. Patients with DMD manifest their first clinical symptoms at 3 to 4 years of age and become wheelchair-dependent between the ages of 7 and 13 years. The ambulation period can be prolonged in many boys with DMD with early initiation of steroid treatment. The terminal stage of the disease begins when patients require assisted ventilation by the age of 20 and patients usually die in their third or fourth decade due to respiratory or cardiac failure[3–7]. DMD results from mutations in the dystrophin gene that cause the synthesis of nonfunctional dystrophin or its absence altogether. Dystrophin is a

[1]Pathophysiology and Genetics of Neuron and Muscle (INMG-PGNM), CNRS UMR 5261, INSERM U 1315, Université de Lyon, Lyon, France. [2]Centre de Biotechnologie Cellulaire, Hospices Civils de Lyon, Lyon, France. [3]Department of Cellular and Molecular Medicine, Faculty of Medicine, 451 Smyth Road, University of Ottawa, Ottawa, ON K1H 8M5, Canada. [4]Éric Poulin Centre for Neuromuscular Disease, Faculty of Medicine, University of Ottawa, Ottawa, ON K1H 8M5, Canada. [5]These authors contributed equally: Bernard J. Jasmin, Laurent Schaeffer. ✉e-mail: alexis.osseni@univ-lyon1.fr; jasmin@uottawa.ca; laurent.schaeffer@univ-lyon1.fr

critical component of the dystrophin-associated glycoprotein complex (DGC) in muscle[8]. DGC is a structure that spans the sarcolemma and forms a mechanical link between the internal cytoskeleton and the extracellular matrix via the association of dystrophin with both actin and microtubule cytoskeletons and the binding of dystroglycan to laminin in the basal lamina, respectively[9,10]. During muscle contraction, the DGC acts as a molecular shock absorber and stabilizes the plasma membrane[11,12]. Loss of dystrophin is associated with loss of the DGC and ensue muscle deterioration and degeneration thereby preventing the DGC to exert its functions. This makes muscle fibers more susceptible to contraction-induced membrane damage leading to cell death[13–16]. This pathologic process is accompanied by inflammation and fibrosis that participate to muscle wasting and loss of function[17–21].

Despite tremendous research efforts, no cure is available for DMD patients yet. Gene-based therapeutic strategies, such as exon skipping, suppression of stop codons, Adeno-associated virus (AAV)-mediated mini-dystrophin delivery, or CRISPR/Cas9 gene editing are actively being investigated to treat DMD[22,23]. In parallel, pharmacological treatments are also being developed[24–27]. Such approaches act on specific signaling pathways and cellular events including those that can cause upregulation of utrophin A[28–30]. Nonetheless, glucocorticoids still serve as the gold standard therapy, acting mostly as anti-inflammatory drugs[31]. With the use of steroids and multidisciplinary care, particularly mechanical ventilation, lifespan expectancy of DMD patients has been considerably extended and affected individuals can now reach 30 to 40 years of age.

TGF-β signaling plays a central role in promoting muscle atrophy and fibrosis in neuromuscular disorders. A variety of ligands including GDF8/myostatin, GDF11 and Activins bind to type II TGF-β receptors and trigger phosphorylation of Smad2 and Smad3 transcription factors upon activation. Smad2/3 phosphorylation enables oligomerization with Smad4 and translocation into the nucleus to activate expression of genes involved in muscle atrophy in cooperation with FoxO transcription factors[32–34]. Inhibition of Smad2 and Smad3 is sufficient to induce muscle growth and constitutive expression of Smad3 triggers muscle wasting[35,36]. Interestingly, previous work has shown that crosstalk exists between TGF-β and mTOR signaling. Indeed, myostatin/GDF8 inhibits Akt/mTOR signaling via a Smad3-induced increase of PTEN translation[37]. Altogether, it appears that TGF-β signaling is an attractive pharmacological target to prevent muscle atrophy. Such drug-based therapies could have significant benefits for atrophic conditions including several neuromuscular diseases, cachexia, and sarcopenia.

HDAC6 is a unique cytoplasmic member of the histone deacetylase (HDAC) family, belonging to class IIb HDAC[38,39]. HDAC6 deacetylates several cytoplasmic substrates such as α-tubulin, cortactin and HSP90, whereas other HDACs are typically central regulators of gene expression. Indeed, classical HDACs such as HDAC4 and HDAC5, are involved in the regulation of atrogenes via Myogenin and Foxo3 transcription factors[40,41]. In this context, we have previously shown that HDAC6 is an atrogene activated by FoxO3a that can interact with the ubiquitin ligase atrogin1/MAFbx[42]. Considerable efforts to develop specific HDAC6 inhibitors have been made to treat a variety of diseases. In particular, several anti-cancer treatments targeting HDAC6 have been proposed[43,44]. Furthermore, HDAC6 deletion or inhibition have been shown to be beneficial for some neurodegenerative disorders, including amyotrophic lateral sclerosis (ALS) and Charcot-Marie-Tooth disease[45–47]. The HDAC inhibitor, tubastatin A (TubA) stands out as a highly selective HDAC6 inhibitor. Indeed, TubA efficiently and specifically inhibits HDAC6 deacetylase activity with an IC50 of 15 nM and a strict selectivity for HDAC6 over all other HDACs (over 1000-fold), except for HDAC8 (57-fold)[46,48,49]. TubA is known to inhibit TGF-β-induced type-1 collagen expression in lung fibroblasts[50,51] and previous reports have shown that TGF-β increases the activity of HDAC6[50,52]. Indeed, Smad7 regulates expression of HDAC6 in prostate cancer in response to TGF-β[53] and HDAC6 may play an indispensable role in balancing the maintenance and activation of primordial follicles through mechanistic target of rapamycin (mTOR) signaling in mice[54,55].

Previous works with pan-HDAC inhibitors have shown that HDAC inhibition improves the dystrophic phenotype[27,56]. However, it currently remains unclear which HDAC is involved in this beneficial effect. In a recent study, we discovered that HDAC6 regulates microtubule stability and acetylcholine receptors (AChR) clustering in normal muscle fibers[57]. We thus wondered whether HDAC6 could play a role in the patho-mechanism of DMD and be used as a novel therapeutic target for the disease. To test this, we investigated the effect of TubA, a highly selective inhibitor of HDAC6, in the DMD *mdx* mouse model and C2C12 cells. Altogether, our results show that pharmacological inhibition of HDAC6 deacetylase activity with TubA is beneficial to muscle from *mdx* mice, both functionally and morphologically. Mechanistic investigation of the mode of action of TubA revealed that HDAC6 inhibition down-regulates TGF-β signaling through an increase of Smad3 acetylation in skeletal muscle.

## Results

### Tubastatin A inhibits HDAC6 deacetylase activity in vivo and improves muscle strength in *mdx* mice

To evaluate the effect of the inhibition of HDAC6 deacetylase activity in dystrophin-deficient mice, 7-week-old *mdx* mice received daily intraperitoneal injections of TubA (*mdx*-TubA) or vehicle (*mdx*-veh) at a dose of 25 mg/kg/day as previously described[46,58]. The mice were treated for 30 days to allow comparison with previous pre-clinical studies using this duration of treatment in *mdx* mice[59–61]. Untreated wild-type, control mice (WT-CTL) mice were used as baseline controls (Fig. 1a). The efficiency of the treatment with TubA was first evaluated by measuring the increase in tubulin acetylation (Fig. 1b). As shown in Fig. 1c, 1d, TubA treatment for 30 consecutive days caused, as expected, a large increase in α-tubulin acetylation in *Tibialis anterior* (TA) muscle from *mdx* mice. In line with published data[57,58], quantification of the relative level of acetylated tubulin showed a -36.5% ± 7.8% increase following treatment with the selective HDAC6 inhibitor ($P < 0.001$ compared with *mdx*-veh mice). To confirm that HDAC6 inhibition did not affect histone acetylation, histone H3 acetylation on lysine 9 (ac-H3K9) and total histone H3 levels in TA muscles were evaluated by Western blot. Consistent with ongoing muscle damage in *mdx* mice, H3K9 acetylation was increased in TA muscles from *mdx* mice compared to WT-CTL animals (+43.8% ± 14.6%; Supplementary Fig. 1a, b). HDAC6 protein level was increased in *mdx* mouse muscles compared to WT-CTL muscles (+97% ± 18.7%; Supplementary Fig. 1c, d). However, TubA increased neither H3K9 acetylation nor HDAC6 levels in *mdx* mice ($P > 0.05$). Together, these data indicate that intraperitoneal injection of TubA efficiently and specifically inhibits HDAC6 deacetylase activity in muscle.

Next, we evaluated the effect of the TubA treatment on muscle strength in *mdx* mice. Seven-week-old untreated *mdx* mice exhibited significantly lower grip strength for all paws ($P < 0.01$) compared with WT-CTL mice (Fig. 1e). After 30 days of treatment with TubA, grip strength of *mdx* mice was significantly increased by 1.5-fold ($P < 0.05$ compared with *mdx*-veh mice; Fig. 1e), with a substantial force gain over 30 day ($P < 0.05$; Fig. 1f). TubA treatment almost completely restored muscle strength of *mdx* mice to WT levels ($P = 0.5476$ compared with WT-CTL mice; Fig. 1e). Interestingly, the maximal force was also increased by 35% ± 9.0% in TubA-treated *mdx* mice compared to *mdx*-veh animal although it did not reach the significance level due to interindividual variability ($P = 0.062$; Fig. 1g). To evaluate muscle fatigability, a serie of 8 successive grip tests at 15 second intervals were performed after 30 days of TubA treatment. Vehicle-treated *mdx* mice showed a progressive decline in strength, indicating fatigue. Treatment with TubA efficiently prevented this drop in grip strength, with a 24.5% ± 6.4% increase grip strength at the height pull compared to

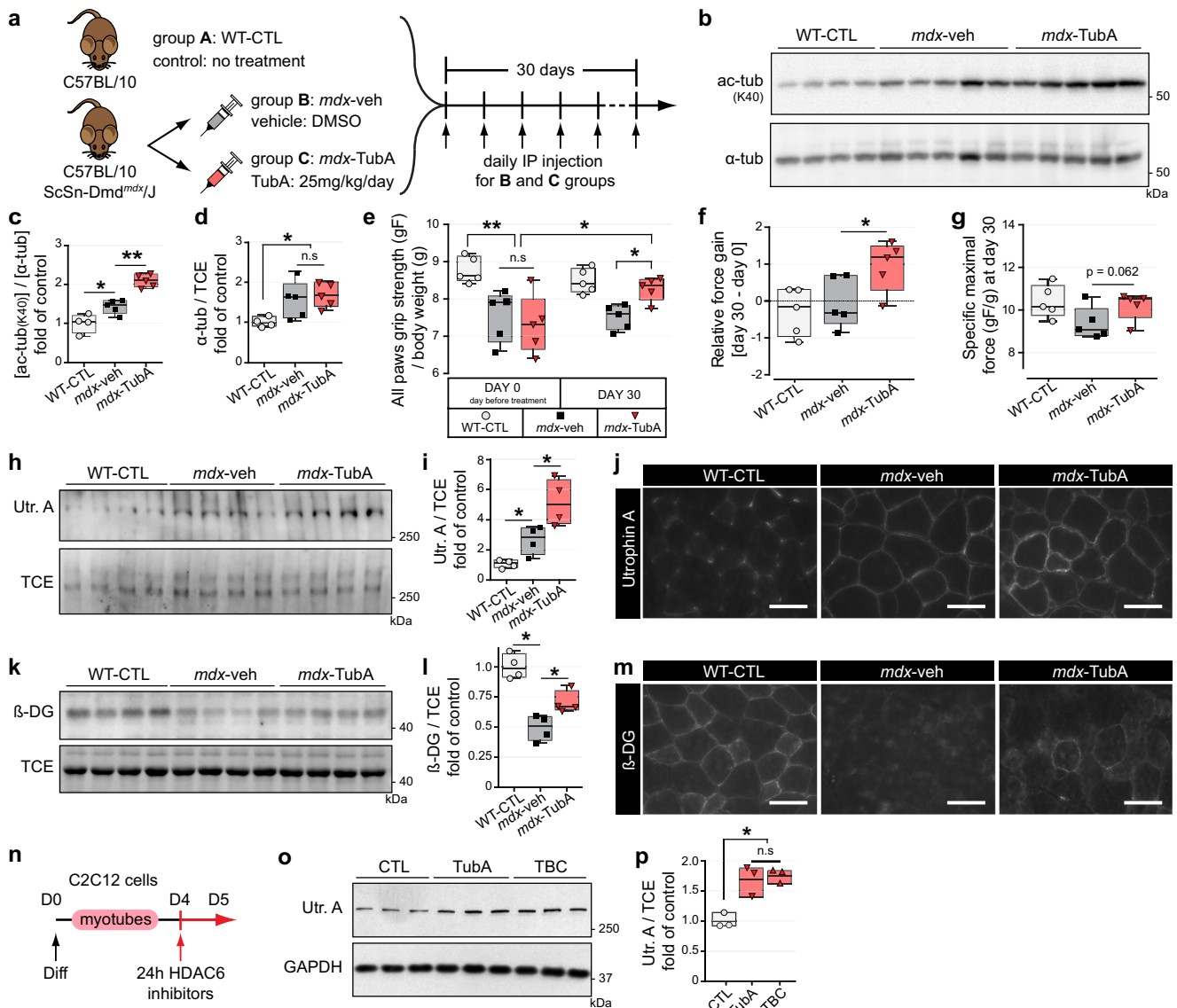

**Fig. 1 | HDAC6 inhibition via TubA treatment increases grip strength, sarcolemmal localization of utrophin A and promotes reassembly of the DGC in _mdx_ mice. a** Protocol of TubA treatment. Three groups of 7-wk-old mice have been evaluated for 4 weeks either without treatment (group A, C57BL/10 mice: WT-CTL), or treated with daily injection for 30 consecutive days with DMSO (group B, C57BL/10ScSn-Dmd_mdx_/J mice; _mdx_-veh) or with TubA at 25 mg/kg/day (group C, C57BL/10ScSn-Dmd_mdx_/J mice; _mdx_-TubA). **b** To evaluate the level of tubulin acetylation (ac-tubK40) in TA muscles, Western blot analysis were performed. Quantifications of acetylated α-tubulin (**c**) and α-tubulin (α-tub, **d**) protein levels ($n = 4-5$ mice per group) were respectively normalized to α-tubulin and 2,2,2-Trichloroethanol (TCE). **e** Grip strength was measured on a grid measuring maximal hindlimb grip strength normalized on body weight ($n = 5$ mice per group). **f** Relative force gain was calculated by the difference between grip strength measured at the last day (day 30) and the day before starting treatment (day 0) and measured with a paired t-test

($n = 5$ mice per group). **g** Specific maximal force was evaluated by the best score of grip strength obtain in each animals at day 30 ($n = 5$ mice per group). To evaluate levels of utrophin A (Utr. A) and β-dystroglycan (β-DG) in TA muscles, Western blot analysis (**h**, **k**) and quantification (**i**, **l**) were performed ($n = 4$ mice per group). TCE was used as a loading control. Cross sections of EDL muscle were stained with an antibody against utrophin A (**j**, in gray) or against β-dystroglycan (**m**, in gray). Scale bars: 50 μm. **n** 4-d-old C2C12 myotubes pretreated for 24 h with different HDAC6 inhibitors TubA (5 μM) and tubacin (TBC, 5 μM) or with DMSO (CTL). **o** Representative Western blots showing utrophin A. GAPDH was used as a loading control. **p** Quantification of Utrophin A protein levels normalized with GAPDH ($n = 3$ independent experiments quantified). (**c**, **d**, **e**, **f**, **g**, **i**, **l**, **p**) Whiskers min to max; the line in the middle of the box is plotted at the median. *$P < 0.05$; **$P < 0.01$; n.s, not significant, $P > 0.05$; Mann–Whitney U test. kDa, relative molecular weight in kiloDalton.

---

vehicle-treated _mdx_ mice ($P < 0.05$; Supplementary Figure 1e). Altogether, these data indicate that inhibition of HDAC6 deacetylase activity in _mdx_ mice restores muscle strength and fatigability to WT levels.

### Tubastatin A treatment increases utrophin and β-dystroglycan levels in dystrophin-deficient mice

Because of its functional and structural similarity with dystrophin, utrophin A can compensate for the lack of dystrophin in DMD[24,29,62,63].

In adult and healthy muscle fibers, utrophin A is exclusively localized at myotendinous junctions and synapses[64,65] and it is now well established that utrophin A expression all along muscle fibers can efficiently compensate for the lack of dystrophin[24,29,66–68]. _Mdx_ mice present an overall milder phenotype than DMD patients[69]. This is partially explained by the compensatory up-regulation of sarcolemmal utrophin A expression[24,25,29,62,63,70,71]. Muscles from all groups of animals described in Fig. 1a were thus analyzed by Western blot and immunofluorescence to evaluate the levels of utrophin A. As expected,

utrophin A levels were increased at the sarcolemma ($P < 0.05$, Fig. 1 h) in *mdx* mice compared with WT-CTL mice. TubA-treatment further increased utrophin A levels by ~2-fold compared with *mdx*-veh mice ($P < 0.05$; Fig. 1i). Immunofluorescence experiments further established that TubA treatment indeed caused an increase in sarcolemmal utrophin A levels in *mdx* mouse muscles, thereby possibly conferring a higher protective effect on muscle fiber integrity (Fig. 1j). Additionally, we assessed both the amount and localization of β-dystroglycan (β-DG), a member of the DGC, in order to determine whether TubA treatment caused reassembly of the DGC along the sarcolemma. As expected, in the absence of dystrophin, β-DG accumulation at the plasma membrane is strongly decreased (Fig. 1k, m). Western blot quantification revealed that TubA increased the amount of β-DG in *mdx* mice by 43.2% ± 7.0% ($P < 0.05$; Fig. 1l). Immunofluorescence experiments with TubA (Fig. 1m) indicated that in addition to the increase in Utrophin A levels, β-DG accumulation at the plasma membrane was increased, suggesting that TubA treatment promoted reassembly of the DGC complex.

To determine whether the effects of TubA on DGC expression was also observed in cultured muscle cells, C2C12 myotubes were treated for 24 hours with 5 μM of one of two selective inhibitors of HDAC6 deacetylase activity: TubA and tubacin[72] (TBC; Fig. 1n). After the treatment, utrophin A expression was evaluated on whole cell extracts by Western blot (Fig. 1o). Both HDAC6 inhibitors induced a significant upregulation of utrophin A levels ~1.75-fold ($P < 0.05$) in C2C12 muscle cells compared to vehicle-treated cells (Fig. 1p). Together, these data demonstrate that treatments with HDAC6 inhibitors increase utrophin A levels in C2C12 myotubes, in agreement with the data obtained in vivo with *mdx* mice.

## HDAC6 inhibition protects from atrophy and reduces muscle damage

To determine whether TubA treatment provided additional benefits to *mdx* muscle fibers, muscle atrophy was evaluated. Previous histo-pathological studies have shown that abnormal fiber size distribution with a strong increase in very small fibers, is a hallmark of dystrophic muscles[73]. Fiber size distribution was thus assessed by measuring the cross-sectional area (CSA) of individual fibers in the slow oxidative *soleus* muscle (SOL) and the fast glycolytic *extensor digitorum longus muscle* (EDL), from *mdx* mice treated with TubA or vehicle and in control mice. As expected, the CSA profiles of vehicle-treated *mdx* muscles displayed substantial heterogeneity and had many small fibers when compared to healthy muscles (Fig. 2a, b). In all muscles analyzed, TubA restored fiber size distribution to a level comparable to that of WT muscles by significantly decreasing the proportion of small muscle fibers and reducing the proportion of large muscle fibers (Fig. 2c, d). In TubA-treated *mdx* mice, these muscle adaptations were accompanied by a significant decrease of the average variance coefficient. Indeed, TubA treatment reduced by ~25% ($P < 0.01$, compared with *mdx*-veh mice) the variance coefficient of fibers CSA in *mdx* EDL and SOL muscles (Fig. 2e, f). Overall, 30 days of treatment with TubA normalized fiber size distribution in both EDL and SOL *mdx* muscles, indicating a protective effect of TubA against muscle atrophy and degeneration.

We previously showed that HDAC6 contributes to muscle atrophy[42]. Given the overall increase in fiber size observed in *mdx* muscles upon TubA treatment, we investigated the mechanism by which pharmacological inhibition of HDAC6 decreased muscle atrophy in *mdx* mice (Fig. 2 g, i). To this end, the expression of protein markers involved in muscle atrophy was evaluated by Western blot. TubA caused a 57% ± 11% reduction of MAFbx/atrogin1 ($P < 0.05$; Fig. 2 h) and a 35% ± 4% reduction of the E3 ubiquitin-protein ligase MuRF1/TRIM63 ($P < 0.05$; Fig. 2j) compared with *mdx*-veh animals. Together, these data indicate that HDAC6 inhibition reduces expression of key mediators of muscle atrophy, thus

providing an explanation for the normalization of the size of muscle fibers.

In addition, heterogeneity in *mdx* fiber size results mainly from the presence of small regenerating fibers subsequent to the loss of necrotic fibers. Central nucleation is as a key indicator of muscle damage in dystrophic muscle fibers[73,74]. To evaluate the effect of TubA on the extent of muscle damage, the number of fibers with central nuclei was evaluated using hematoxylin eosin staining (Fig. 2k). TubA caused a significant decrease in the number of centronucleated fibers (−15% ± 2% in EDL muscle, Fig. 2 l and −24% ± 2% in SOL muscle, Fig. 2 m; both $P < 0.05$, compared with *mdx*-veh mice). Furthermore, the loss of muscle fibers is accompanied by the progressive accumulation of fibrotic tissue[75]. We therefore investigated by Western blot if the expression of fibrosis-associated proteins in *mdx* muscles was reduced by TubA (Fig. 2n, p). The protein levels of type I Collagen Alpha 1 Chain (Col1A1) and connective tissue growth factor (CTGF) were both reduced in *mdx* mice treated with TubA compared to vehicle treated *mdx* mice (−36% ± 7%, Fig. 2o and −16% ± 1%, Fig. 2q; respectively, $P < 0.05$). To confirm these results, Masson's trichrome staining was performed on cryosections of SOL (Fig. 2r) and EDL muscles (Supplementary Fig. 2a). TubA significantly decreased fibrotic area in *mdx* SOL muscle ($P < 0.001$; Fig. 2s) compared to control *mdx* SOL (vehicle treated). In EDL muscle, TubA effect on the overall collagen positive surface did not reach significance (P > 0.05; Supplementary Figure 2b). Nevertheless, TubA efficiently prevented the formation of areas with high collagen content in *mdx* EDL muscle. Altogether, these data show that HDAC6 inhibition in dystrophic muscle reduces central nucleation, normalizes fiber size distribution, decreases the proportion of small fibers, and prevents fibrosis.

## HDAC6 inhibition restores microtubule network and neuromuscular junction organization in *mdx* mice

Dystrophin and utrophin A link the DGC complex to the microtubule network[10,76]. In agreement with previous data[10,77], we confirmed by Western blotting and immunofluorescence staining that *mdx* mouse TA and EDL muscle fibers have more tubulin and a greater microtubule density, respectively, than WT-CTL mice ($P < 0.05$, Fig. 1d and Supplementary Fig. Fig. 3a, b). However, TubA treatment did not affect the overall α-tubulin abundance ($P > 0.05$ compared with *mdx*-veh mice; Fig. 1d). In healthy muscle, the microtubule network forms a grid lattice with longitudinal, transverse, and perinuclear microtubules[78–80]. Here, in WT-CTL mice, we observed that transverse and longitudinal microtubules are regularly spaced by ~2 μm and ~5 μm, respectively (see arrowheads Fig. 3a, and Supplementary Fig. 3c–e). In *mdx* muscles, immunofluorescence experiments show a disorganization of the microtubule network, with a loss of the grid-like organization (see arrows Fig. 3a). Interestingly, in *mdx* mice treated with TubA the microtubule lattice was restored (see the spacing between microtubules in the blue and yellow line scans and Supplementary Fig. 3c–e). We further analyzed the microtubule network with a software specifically developed to analyze microtubule directionality[81] (TeDT program; Fig. 3b). TeDT program analysis revealed a significant decrease of transversely oriented microtubules (centered around 90°) in vehicle-treated *mdx* muscles compared to WT and to TubA-treated *mdx* mice ($P < 0.05$ compared with *mdx*-veh mice). These data show that TubA treatment improves the organization of the microtubule network in *mdx* muscles. In agreement with previous work[57,82], this further demonstrates that microtubule acetylation via HDAC6 inhibition stabilizes microtubules and favors their transverse orientation.

The highly fragmented distribution of post-synaptic acetylcholine receptor (AChR) observed in *mdx* muscles strongly suggests that dystrophin participates to the maintenance of the NMJ[83]. Since HDAC6 inhibition increases the compactness of AChR distribution at the NMJ[57], we analyzed NMJ organization in control and treated *mdx* mice (Fig. 3c). α-bungarotoxin staining, used to visualize the acetylcholine

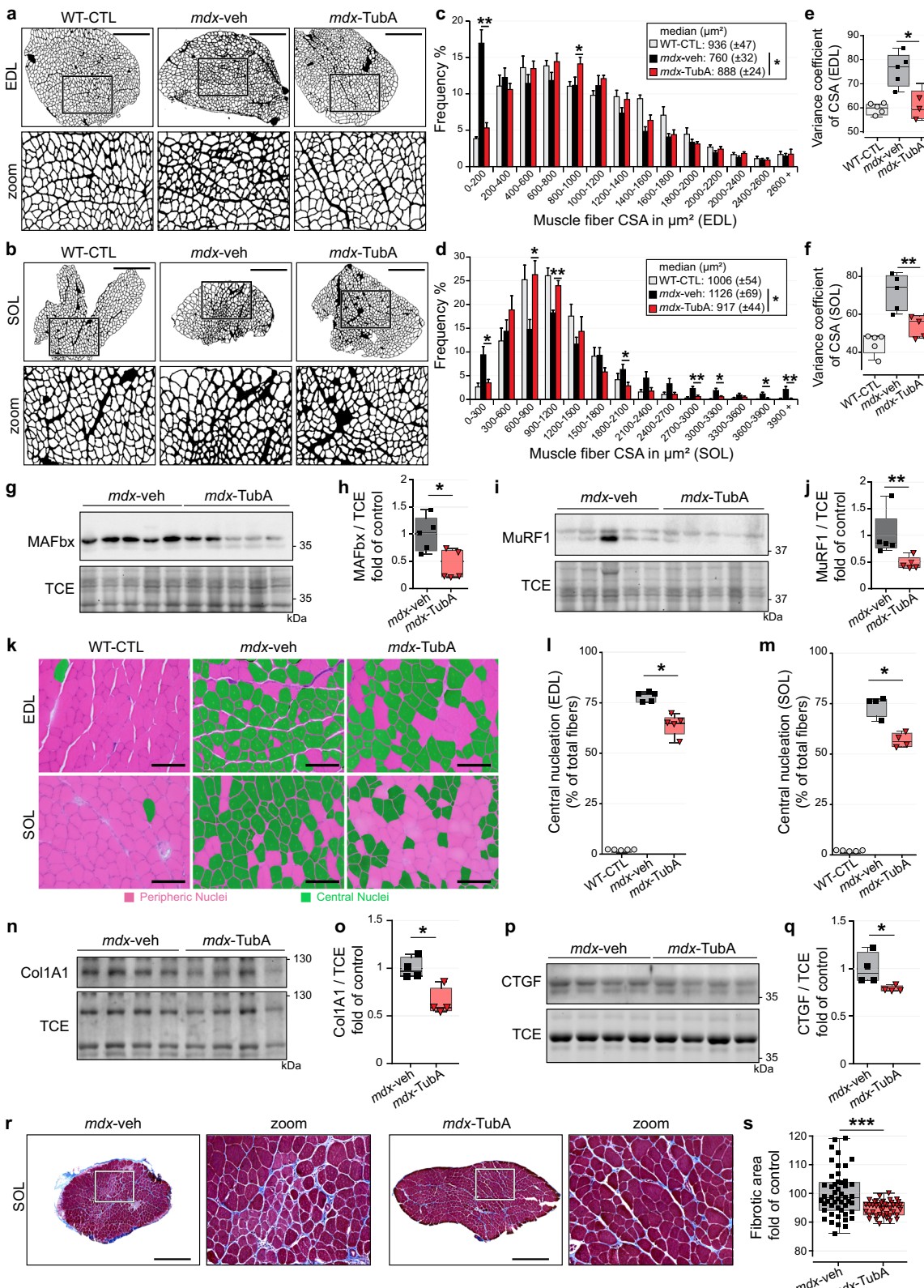

receptor, indicated that NMJ organization was severely compromised in control *mdx*-veh treated mice as compared to control WT NMJs (Fig. 3c, d), as expected from previous works[84–86]. More specifically and compared to WT muscles, the compactness of *mdx*-veh NMJs was decreased by 26% ± 11% (*P* < 0.05; Fig. 3e), the overall surface of NMJs was increased by 38% ± 18% (*P* < 0.05; Fig. 3d), the fragmentation index

was increased by approximately twofold (*P* < 0.001; Fig. 3g) and the average number of fragments per NMJ was increased (*P* < 0.001; Fig. 3f). TubA treatment restored almost to normal the NMJ surface (−24% ± 13%, *P* < 0.05 compared to vehicle treated *mdx* NMJs; Fig. 3d) and compactness (+31.3% ± 12.5%, *P* < 0.05 compared to vehicle treated *mdx* NMJs; Fig. 3e and Supplementary Fig. 3f, g). Compared to vehicle-

**Fig. 2 | TubA treatment improves and restores DMD phenotype in *mdx* muscle and protects from atrophy.** Cross-section areas (CSA) of entire EDL (**a**) and SOL (**b**) muscles from 11-wk-old C57BL/10 mice (WT-CTL) and C57BL/10ScSn-Dmd*mdx*/J mice treated with vehicle-DMSO (*mdx*-veh) or with TubA (*mdx*-TubA) for 30 consecutive days were stained using laminin staining and then binarized on ImageJ (*n* = 5 mice per group). Scale bars: 500 μm. Graphical summary of CSA in EDL (**c**) and SOL (**d**) muscle (n = 5 mice per group; entire EDL and SOL muscles were counted per mouse). **c**, **d** Data are presented as mean values ± SEM. *P < 0.05; **P < 0.01; two-way ANOVA (*mdx*-TubA versus *mdx*-veh). Median CSAs of each muscle are displayed above the frequency histograms. Measurements of variance coefficient in EDL (**e**) and SOL (**f**) muscle fibers (*n* = 5 mice per group). To evaluate levels of MAFbx (**g**, **h**) and MuRF1 (**i**, **j**) in TA muscles, Western blot analysis (**g**, **i**) and quantifications (**h**, **j**) were performed (*n* = 5 mice per group). **k** Representative

examples of cross-sections of EDL and SOL muscles were stained using hematoxylin and eosin. Scale bars: 100 μm. Centrally nucleated fibers are colored in green. Percentage of central nucleation in EDL (**l**) and SOL (**m**) muscle fibers (*n* = 4 or 5 mice per group). To evaluate levels of collagen type I alpha 1 (**n**, **o**, Col1A1) and connective tissue growth factor (**p**, **q**, CTGF) in TA muscles, Western blot analysis (**n**, **p**) and quantifications (**o**, **q**) were performed (n=4 mice per group). TCE was used as a loading control for all Western blots. To evaluate level of collagen content infiltration, Masson's trichrome staining (**r**) and quantification (**s**) were performed in SOL muscle. Scale bars: 500 μm. Fibrotic area are colored in blue (*n* = 28–50 fields counted per mouse, 3 mice per group). (**e**, **f**, **h**, **j**, **l**, **m**, **o**, **q**, **s**) Whiskers min to max; the line in the middle of the box is plotted at the median. *P < 0.05; **P < 0.01; ***P < 0.001; n.s, not significant, P > 0.05; Mann–Whitney U test. kDa, relative molecular weight in kiloDalton.

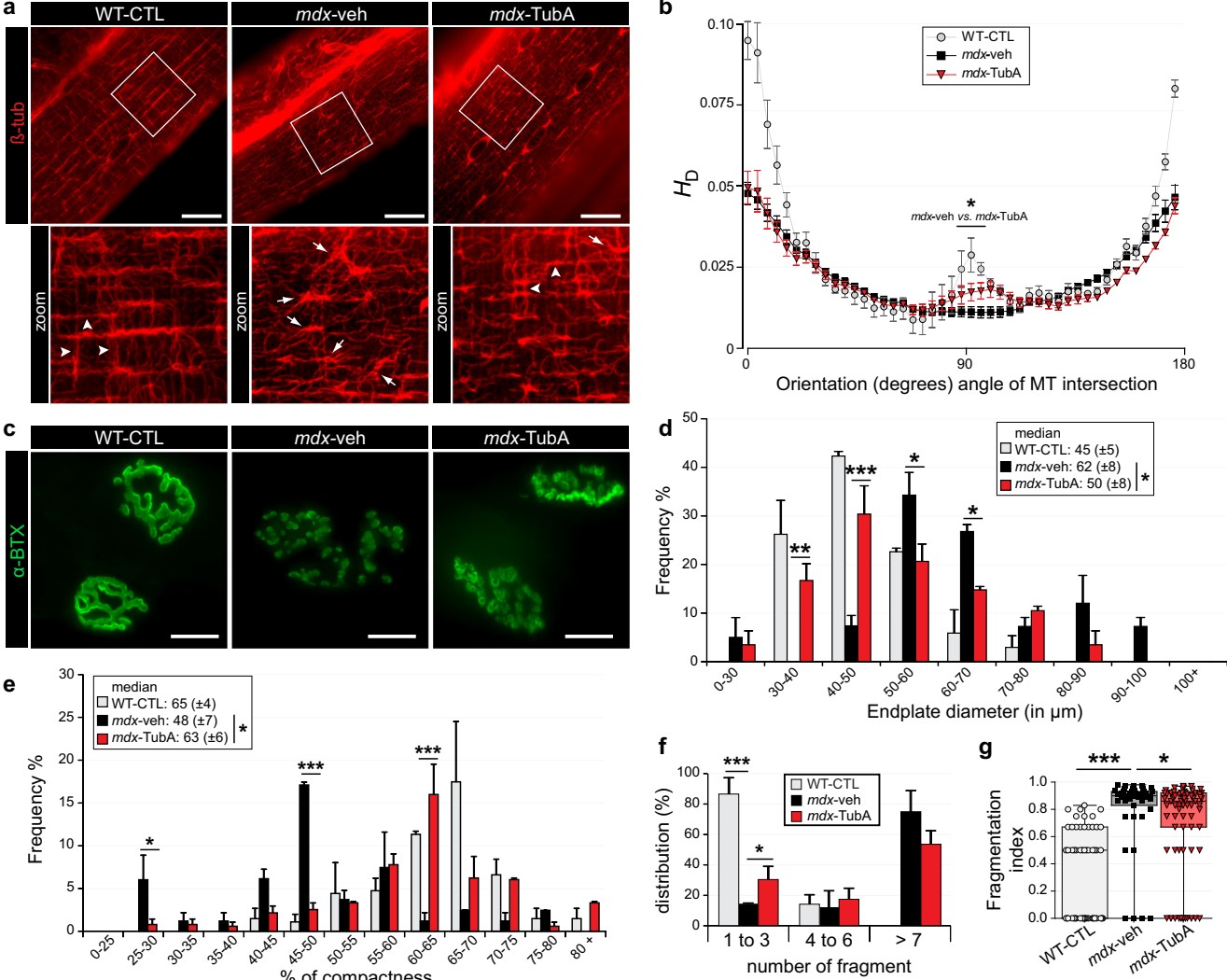

**Fig. 3 | TubA treatment stabilizes MT network and protects NMJ morphological characteristics from dystrophic mice. a**, **c** Isolated fibers of TA from 11-wk-old C57BL/10 mice (WT-CTL) and C57BL/10ScSn-Dmd*mdx*/J mice treated with vehicle-DMSO (*mdx*-veh) or with TubA (*mdx*-TubA) for 30 consecutive days were stained with an antibody against β-tubulin (ß-tub) to label MT network (**a**, in red) or stained with α-bungarotoxin-A488 (**c**, in green, α-BTX–A488) to label NMJs. Scale bars: 25 μm. **b** MT network organization analysis using TeDT software (*n* = 4 to 6 fibers from 3 mice). The final generated graph presents a global score for each given degree of MT orientation, with 0 and 180 degrees corresponding to the longitudinal MT and 90 degrees corresponding to the transverse MT; directional

histograms (HD). Arrowheads represent regular organization of MT network whereas arrows show disorganization of the MT network, with a loss of the grid-like organization. Graphical summary of NMJ endplate diameter (**d**) and NMJ compactness (**e**) were performed. **d**, **e**, *, ** and *** comparison between *mdx*-veh vs. *mdx*-TubA mice. Median of each group of mice are displayed above the frequency histograms (*n* = 40–75 NMJs counted). Distribution of number of fragments (**f**) and fragmentation index (**g**) have been quantified (n = 43-73 of NMJs counted). **g** Whiskers min to max; the line in the middle of the box is plotted at the median. *P < 0.05; **P < 0.01; ***P < 0.001; (*mdx*-TubA versus *mdx*-veh), Mann–Whitney U test.

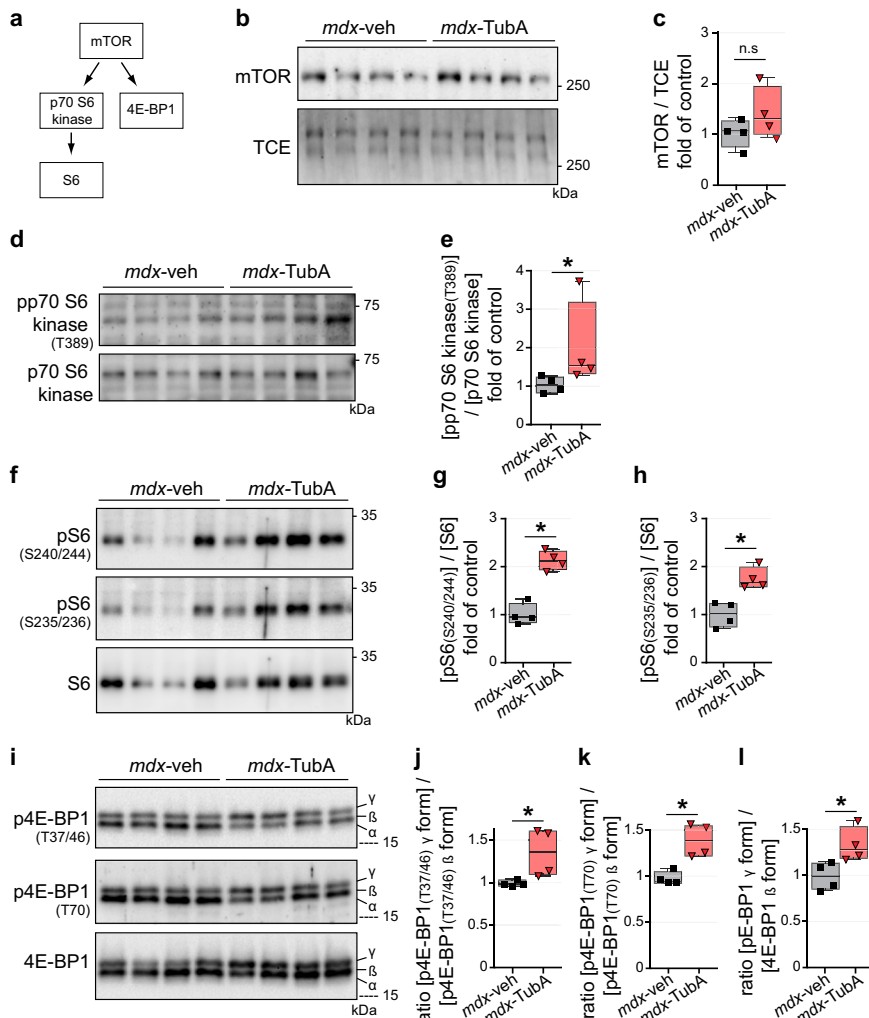

**Fig. 4 | HDAC6 inhibition actives mTOR pathway in *mdx* mice. a** Schematic summary of downstream targets of mTOR. 11-wk-old C57BL/10ScSn-Dmd*mdx*/J mice treated with TubA (*mdx*-TubA) or with vehicle-DMSO (*mdx*-veh) for 30 consecutive days have been analyzed in TA muscle by Western blot analysis (**b**, **d**, **f**, **i**). Quantifications of mTOR (**c**), pp70 S6 kinase (**e**, T389), pS6 (**g**, S240/244), pS6 (**h**, S235/236), p4E-BP1 (**j**, T37/46), p4E-BP1 (**k**, T70), and 4E-BP1 (**l**) were performed (*n* = 4 mice per group). TCE was used as a loading control. (**c**, **e**, **g**, **h**, **j**, **k**, **l**) Whiskers min to max; the line in the middle of the box is plotted at the median. *\**P* < 0.05; n.s, not significant, *P* > 0.05; Mann–Whitney U test. kDa, relative molecular weight in kiloDalton.

treated *mdx* mice, TubA also partially normalized the fragmentation index −13.2% ± 2% (*P* < 0.05; Fig. 3g) and the number of fragments per NMJ (+63% ±1% of NMJs composed of 1 to 3 fragments, *P* < 0.05; Fig. 3f). Altogether, TubA-treatment partially restored normal AChR distribution in *mdx* mouse muscle fibers via an increase in compactness and recruitment of AChR patches.

**Inhibition of HDAC6 activates protein synthesis signaling via mTOR pathway**

Dystrophin transcription is known to also be controlled via the mTOR pathway[87]. Indeed, mTOR deficiency leads to reduced muscle dystrophin content and causes dystrophic defects leading to severe myopathy. In this context, we investigated if HDAC6 inhibition in *mdx* muscle could promote protein synthesis by measuring phosphorylation of various components of the mTOR pathway (Fig. 4a). mTOR protein level was not significantly changed with TubA treatment (*P* = 0.342; Fig. 4b, c) whereas a ~2-fold increase of p70-S6 kinase phosphorylation at threonine 389 was observed (*P* < 0.05; Fig. 4d, e). Consistent with increased p70-S6 kinase activity, we observed a ~2-fold increase of the phosphorylation of S6 on both Ser235-236 and Ser240-244 (*P* < 0.05; Fig. 4f, g, h). Finally, TubA induced a significant increase in the phosphorylation of the translation inhibitor 4E-BP1 on

the mTOR-sensitive sites as shown by the increase in the ratio of 4E-BP1 γ over 4E-BP1 β isoforms for phospho-Thr37-46, phospho-Thr70 as well as for total 4E-BP1 (Fig. 4i, j, k, l). Altogether, our data show that a 30 day-treatment with TubA stimulates muscle protein synthesis in *mdx* mice via the mTOR pathway.

To determine whether the effect of TubA on the mTOR pathway was also observed in cultured muscle cells, C2C12 myotubes were pretreated for 24 hours with TubA and then treated with rapamycin, a selective inhibitor of mTOR complex 1 (mTORC1) (Supplementary Fig. 4a). After the treatment, phosphorylation of S6 protein was evaluated on whole cell extracts by Western blot (Supplementary Fig. 4b). As expected, based on the in vivo data, treatment with TubA increased S6 phosphorylation in C2C12 muscle cells, compared to vehicle-treated cells (Supplementary Fig. 4c, d). In the presence of rapamycin, S6 phosphorylation was drastically reduced, both in the presence or absence of TubA (*P* < 0.001; Supplementary Fig. 4c, d). Therefore, HDAC6 inhibition acts upstream of the mTOR complex to activate the pathway. In these experiments, we also evaluated if the increase in utrophin A expression by TubA treatment relied on the mTOR pathway. Utrophin A expression was evaluated in the presence of TubA with or without rapamycin. Rapamycin prevented the upregulation of utrophin A by TubA, indicating that the increase in utrophin A induced

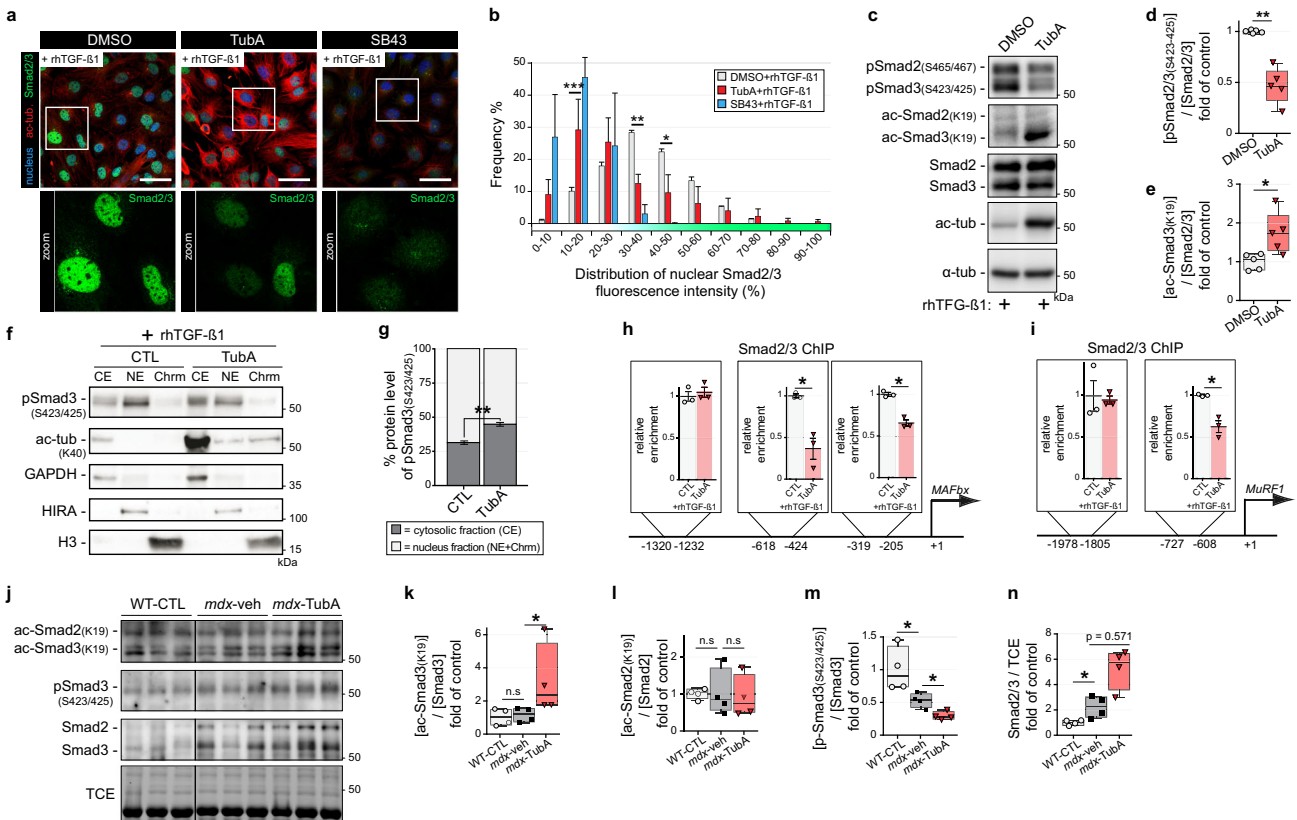

**Fig. 5 | TGF-β signaling is regulated by TubA via acetylation of Smad3. a–e** 4-d-old C2C12 myoblasts pretreated for 24 h with either HDAC6 inhibitor (TubA, 5 μM), or selective inhibitor of TGF-β1 (SB43, 5 μM) or with DMSO (CTL). Myoblasts were then treated for 30 min with recombinant human TGF-β1 (rhTGF-β1; 10 ng/mL). **a** Myoblasts were double-stained with antibodies against Smad2/3 (in green) and acetylated tubulin (ac-tub, in red). Nuclei were labeled with DAPI (in blue). Scale bars: 50 μm. **b** Graphical summary of nuclear distribution of Smad2/3 fluorescence intensity from 3 independent experiments; two-way ANOVA (TubA+rhTGF-β1 versus DMSO + rhTGF-β1). **c** Levels of Smad2/3 phosphorylation (pSmad), Smad2/3 acetylation (ac-Smad), Smad2/3, acetylated α-tubulin (ac-tub), and α-tubulin (α-tub) were visualized by Western blot analysis. To evaluate levels of Smad2/3 phosphorylations (**d**) and Smad3 acetylation (**e**) in C2C12 cells quantifications have been performed (*n* = 5 independent experiments quantified); Mann–Whitney U test. **f, g** Cellular fractionation into 3 fractions was performed: cytosolic fraction (CE), nuclear extract (NE), and chromatin extract (Chrm). **f** Levels of Smad3 phosphorylation (S423-425), Smad2/3, acetylated α-tubulin (ac-tub-K40), GAPDH, HIRA

and histone H3 (H3) were visualized by Western blot from 3 independent experiments. **g** Quantifications of distribution of Smad3 phosphorylation (S423-425); two-way ANOVA. Chromatin immunoprecipitations (ChIP) was performed using antibodies against Smad2/3. qPCR amplifications of immunoprecipitated promoter regions of the MAFbx (**h**) or MuRF1 (**i**) gene were used to detect the presence of these DNA fragments in immunoprecipitates (**h, i**, *n* = 3 independent experiments); Mann–Whitney U test. *, *P* < 0.05. (**j, k, l, m, n**) 11-wk-old C57BL/10ScSn-Dmd*mdx*/J mice treated with vehicle-DMSO (*mdx*-veh) or with TubA (*mdx*-TubA) for 30 consecutive days have been analyzed in TA muscle by Western blot analysis (**j**) and quantified (**k, l, m, n**). To evaluate levels of Smad3 acetylation (**k**), Smad2 acetylation (**l**), Smad3 phosphorylation (**m**) and Smad2/3 (**n**) in TA muscles, quantifications have been performed (*n* = 4 mice per group); Mann–Whitney U test. TCE was used as a loading control for all Western blots. (**d, e, k, l, m, n**) Whiskers min to max; the line in the middle of the box is plotted at the median. (**b, g, h, i**) Data are presented as mean values ± SEM. *P < 0.05; **P < 0.01; ***P < 0.001; n.s, not significant, P > 0.05. kDa, relative molecular weight in kiloDalton.

by TubA involves the mTOR pathway (Supplementary Figure 4f). Both mTOR and TGF-β signaling are key players in muscle mass regulation and these pathways are interconnected[37,88].

### Increased Smad3 acetylation via HDAC6 inhibition protects against TGF-β treatment and Smad2/3 phosphorylation

The effects of TubA on the mTOR pathway, muscle atrophy and fibrosis prompted us to explore if HDAC6 inhibition affected TGF-β signaling. Muscle atrophy induced by the TGF-β member myostatin involves Smad2/3 signaling that activates MAFbx expression and downregulates mTOR signaling[89,90]. Smad2/3 proteins require phosphorylation to enter inside nucleus and activate its target genes[91]. Interestingly, Smad2/3 proteins have also been shown to be acetylated in the nucleus[92,93]. Mouse C2C12 myoblasts were treated for 24 hours with either vehicle (DMSO), TubA or SB431542 (SB43) a specific inhibitor of the TGF-β/Activin/NODAL pathway[94]. After removal of the inhibitors, C2C12 myoblasts were exposed for 30 min to recombinant human TGF-β1[95] (rhTGF-β1) and the subcellular localization of Smad2/3

was analyzed by immunofluorescence using anti-acetylated tubulin and anti-Smad2/3 antibodies.

TubA treatment similarly increased tubulin acetylation both in the presence and absence of rhTGF-β1, indicating that TGF-β signaling does not affect HDAC6 deacetylase activity (Fig. 5a and Supplementary Fig. 3a, c). As expected[96,97], rhTGF-β1 treatment increased Smad2/3 phosphorylation at serines 465-467 and 423-425, respectively (Supplementary Fig. 3b). Consistently, immunofluorescence and Western blot experiments in C2C12 cells, respectively, indicated that rhTGF-β1 increased phosphorylation and nuclear accumulation of Smad2/3 which were blocked by SB43 (Fig. 5a, b and Supplementary Fig. 5a–d). In the presence of TubA, Smad2/3 phosphorylation and nuclear accumulation were strongly diminished (Figs. 5a–d, f–h and Supplementary Fig. 5a–d), and these reduction correlated with an increase in the acetylation of Smad3 (*P* < 0.05, Fig. 5e). To confirm these results, cell fractionation was performed in the presence or absence of TubA to separate a cytosolic fraction (CE) and a nuclear fraction with the latter being further sub-divided into a nuclear extract (NE) and a chromatin

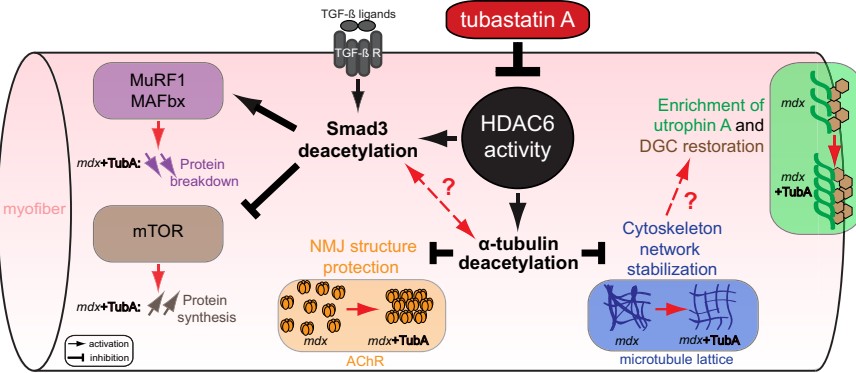

**Fig. 6 | Consequences of HDAC6 activity inhibition regulation through TubA treatment in DMD mice model.** HDAC6 inhibitor such as TubA induces a decrease in HDAC6 activity that leads to an acetylation of α-tubulin and of Smad3. HDAC6 pharmacological inhibition allows an increase of α-tubulin acetylation to restore DGC and stabilize MT network/NMJ organization. Additionally, specific inhibition of HDAC6 increases acetylation of Smad3 which can interfere with TGF-β signaling to both reduce muscle atrophy by reducing the expression of key actors such as MAFbx/MuRF1 and to stimulate protein synthesis via mTOR pathway. Our results identify HDAC6 as a pharmacological target of interest for DMD.

extract (Chrm). In the presence of TubA, the cytoplasmic to ratio of nuclear phosphoSmad3 to cytoplasmic phosphoSmad3 was inverted, confirming that HDAC6 inhibition reduced Smad3 nuclear translocation (Fig. 5f, g).

To investigate the functional impact of this cellular redistribution and determine whether TubA affects activation of Smad2/3 target genes, a Smad2/3 ChIP assay was performed on C2C12 cells treated with rhTGF-β1 or rhTGF-β1 plus TubA. Putative Smad binding sites were identified by software analysis upstream of the transcription start site in the promoters of MAFbx and MuRF1, i.e., between positions −618 and −424 and positions −319 and −205 for MAFbx, and between positions −727 and −608 for MURF1. The −1978 to −1805 and −1320 to −1232 promoter regions were used as negative controls for MAFbx and MURF1, respectively. The presence of TubA significantly reduced the binding of Smad2/3 to the promoters of MAFbx (−63% ± 12% and −34% ± 4%) and MURF1 (−38% ± 7%) at their predicted binding regions ($P < 0.05$; Fig. 5h, i), which nicely complements the cellular fractionation data.

TGF-β signaling is known to be upregulated in dystrophic muscles[98]. To confirm in vivo that HDAC6 inhibition downregulated TGF-β signaling, Smad2/3 acetylation and phosphorylation were evaluated by Western blot in TA muscles of *mdx* mice treated with vehicle-DMSO or TubA (Fig. 5j). In TubA-treated *mdx* mice, Smad3 acetylation on Lys19 was almost threefolds higher than in DMSO-treated *mdx* mice ($P < 0.05$; Fig. 5k), whereas Smad2 acetylation was unaffected ($P > 0.05$; Fig. 5l). This increase in Smad3 acetylation was paralleled by a 55% ± 4% decrease of Smad3 phosphorylation on Ser432-425 ($P < 0.05$; Fig. 5m). In *mdx* mice, TubA treatment had a tendency to increase the total level of Smad2/3 compared to *mdx*-veh animal. However, this increase was not statistically significant ($P = 0.0571$; Fig. 5n). The effect of TubA on Smad4 and Smad7 levels was also evaluated, and no difference was observed with controls (Supplementary Fig. 5e–h).

The mechanisms linking the mTOR and TGF-β/Smad pathways are still poorly understood. Nevertheless, Smad3 was shown previously to activate translation of PTEN mRNAs, thereby inhibiting Akt/mTOR signaling and protein synthesis[39] (Supplementary Fig. 5i). We therefore investigated a possible downregulation of PTEN expression consecutive to Smad 3 inhibition by TubA. Both in *mdx* mouse muscles and in C2C12 cells, TubA induced a decrease in PTEN protein levels ($P < 0.05$; Supplementary Fig. 5j–m). Akt protein levels and phosphorylation were also examined and consistent with the decrease in PTEN levels, both Akt1 and Akt2 protein levels and phosphorylation were increased in the presence of TubA ($P < 0.05$; Supplementary Fig. 5n–s). Altogether this indicated that TubA most likely activates the mTOR pathway via Smad3 and PTEN.

## Discussion

Here, we have evaluated the effect of HDAC6 inhibition by TubA in *mdx* mice and in C2C12 muscle cells. TubA treatment significantly ameliorated *mdx* muscle function and decreased the overall histopathological dystrophic features. In search of the mechanism of action of HDAC6, we discovered that HDAC6 regulated TGF-β signaling via the acetylation of Smad3, thus identifying Smad3 as a new target of HDAC6. TubA increased Smad3 acetylation, preventing Smad2/3 phosphorylation and nuclear translocation, thereby decreasing the expression of atrogenes such as MAFbx and MuRF1 and upregulating the mTOR pathway. Altogether, this mechanism can explain the beneficial effect of HDAC6 inhibition to counter muscle atrophy and fibrosis in *mdx* muscles (Fig. 6).

HDAC6 is widely expressed across tissues in the body[38]. In cancer, HDAC6 is overexpressed and known to correlate with poor prognosis[99,100]. Here, we observed that HDAC6 expression is increased in *mdx* mice, which indicates that it could participate in the DMD pathology. In agreement with this notion, selective inhibition of HDAC6 with TubA diminished multiple pathological features of *mdx* mouse muscle, without affecting the expression levels of HDAC6. This suggests that the beneficial effect of HDAC6 inhibition resides in the reduction of its deacetylase activity, independently of its protein level.

Our results indicate that stabilizing microtubules by preventing their deacetylation preserves microtubule organization in dystrophin-deficient muscles. In *mdx* muscles, the loss of force production is commonly attributed to structural impairments of the muscle fiber cytoskeleton and changes in signaling[101]. Interestingly, stabilization of the microtubule network was shown to protect against contraction-induced injury, suggesting that targeting the microtubule cytoskeleton may provide novel opportunities for therapeutic intervention in DMD[77]. Accordingly, *mdx* muscles treated with TubA contain less centronucleated fibers, possibly because muscle fibers are more resistant.

Our data show that TubA partially restores NMJ morphology in *mdx* mice. In DMD patients and *mdx* mice, NMJs are noticeably disorganized and display deficits in neuromuscular function / transmission, thus highlighting the contribution of NMJ impairment in altered function and recovery of dystrophic muscles[83,102–104]. We recently showed that HDAC6 is involved in microtubule organization at the NMJ, by regulating AChR distribution and NMJ organization[57]. In *mdx* mice, structural changes in microtubules at the NMJ are probably involved in the disorganization of the NMJ[85]. Our present study indicates that protecting NMJ structure by stabilizing microtubules probably participates to the beneficial effect of TubA treatment on *mdx* muscle function. In *mdx* muscles, NMJs are abnormally large because

of fragmentation and our current results indicate that stabilizing the microtubule network limits this fragmentation, which is consistent with our previous results that showed that increasing microtubule acetylation restricts spreading of the NMJ[58]. The increase in utrophin A levels could also participate to the improvement of NMJ organization by TubA.

A significant increase of utrophin A levels correlates with improved prognosis in DMD patients[105]. Therefore, the ability of HDAC6 inhibition to increase extra-synaptic utrophin A levels probably participates to the preservation of *mdx* muscle integrity. How HDAC6 regulates utrophin A remains to be determined yet, we have observed here that rapamycin prevents the increase in utrophin A expression by TubA, indicating the involvement of the mTOR pathway. An additional beneficial action of TubA could also be mediated by the restoration of the microtubule network which, in turn, could improve utrophin A trafficking to the membrane.

In contrast to most HDACs that directly act on gene expression via transcription regulatory complexes, HDAC6 is strictly cytoplasmic and none of its known substrates are transcription factors. Here, we show that HDAC6 deacetylates Smad3 and regulates its nuclear accumulation. Smad2/3 proteins mediate the action of TGF-β signaling to promote protein catabolism and fibrosis and to inhibit protein anabolism. Inhibition of Smad2/3 nuclear accumulation by HDAC6 inhibition can therefore explain the beneficial effect of TubA on renal fibrosis[106], muscle atrophy reduction and protein synthesis stimulation via the mTOR pathway. Beneficial effects of HDAC inhibitors such as Givinostat, were previously demonstrated in *mdx* mice and DMD patients to stimulate the expression of the activin binding protein follistatin, whose main activity is to block TGF-β signaling[27,56,107]. These beneficial effects of HDAC6 on *mdx* muscles recapitulate effects of Givinostat. Moreover, in *mdx* and DMD patient muscles, inhibition of TGF-β activity attenuates both degeneration and fibro-calcification[108] and fibrosis[27]. Consistently, HDAC6 inhibition also reduced fibrosis in *mdx* muscles.

In an attempt to link the effect of TubA on TGF-β signaling to mTOR, we observed that PTEN expression was decreased by TubA, pointing to the known role of Smad3 in the activation of PTEN translation[37]. PTEN levels are known to be elevated in dystrophic muscles of DMD patients and *mdx* mice, and PTEN inhibition was shown to improve muscle regeneration and function in *mdx* mice[109]. Therefore, the effect of TubA on PTEN expression likely participates to the improvement of *mdx* muscles.

Interestingly, rather than directly affecting the expression of individual genes as shown for other HDACs, HDAC6 acts on TGF-β signaling by targeting Smad3 in the cytoplasm. Hence, HDAC6 regulates the downstream targets of Smad2/3, thus providing a novel pharmacological entry to interfere with TGF-β signaling. Smad2 and Smad3 share 92% sequence identity, but their functions are not completely redundant. Smad2 knock-out mice die at embryonic day 10.5 with vascular and cranial abnormalities and impaired left-right patterning[110,111] whereas Smad3 knock-out mice are *via*ble but suffer from impaired immune function and chronic inflammation[112,113]. In addition, they exhibit preferences for association with specific transcription factors. For example, Smad3 interacts preferentially with FoxO over Smad2[32]. Interestingly, previous reports on Smad2/3 acetylation exclusively involved nuclear factors[92,93] and the effect of Smad2/3 acetylation was correlated with promoter binding, transactivation activity, nuclear export or protein stability. In the nucleus, Smad3 is acetylated at lysine 19 by p300/CBP in response to TGF-β, which increases its DNA binding activity and binding to target gene promoters[114]. Acetylation of Smad3 may selectively up- or down-regulate the expression of TGF-β-regulated genes. Of note, Smad3 can also be acetylated on several other lysines[92,93]. HDAC6 being strictly cytoplasmic, Smad3 deacetylation by HDAC6 probably affects different processes. Our results indicate that increasing Smad3 acetylation

reduces the phosphorylation of Smad 3 and its nuclear accumulation. We can thus speculate that preventing Smad3 deacetylation in the cytoplasm reduces its phosphorylation by TGF-β receptors and/or reduce its association with Smad4. Altogether, Smad3 acetylation in the nucleus is required to activate the expression of its target genes, whereas Smad3 acetylation in the cytoplasm inhibits its function. Intriguingly, despite the high level of similarity between Smad2 and Smad3, HDAC6 inhibition selectively increased Smad3 acetylation yet, it affected the phosphorylation and nuclear entry of both Smad2 and Smad3. We hypothesize that acetylated Smad3 in the cytoplasm could bind Smad2 and inhibit its phosphorylation, and that formation of an acetylated Smad2/3 dimer could reduce their ability to enter the nucleus. It is conceivable that after its activation by p300, acetylated Smad2/3 molecules that are exported back from the nucleus to the cytoplasm require deacetylation by HDAC6 to allow their re-entry in the TGF-β pathway. Therefore, HDAC6 inhibition would prevent Smad3 deacetylation in the cytoplasm thereby reducing the amount of Smad2/3 available to participate to TGF-β signaling and increasing the amount of cytoplasmic Smad2/3. Another possibility would be that Smad3 deacetylation by HDAC6 regulates its ubiquitination and subsequent degradation. Both hypothesis are consistent with the fact that HDAC6 inhibition increases the amount of cytoplasmic Smad2/3 protein.

Functional connections between HDAC6 and TGF-β were previously reported. Indeed, Smad3 phosphorylation and nuclear accumulation was shown to be regulated by HDAC6[106,115]. TGF-β was also shown to induce α-tubulin deacetylation by HDAC6, resulting in increased cell motility[115], and microtubule stability regulates TGF-β-induced Smad phosphorylation[116–118]. By showing here that HDAC6 regulates both Smad3 and microtubules acetylation, our results provide a direct molecular link between microtubules stability and TGF-β signaling.

In a previous work, we showed that HDAC6 expression was increased during muscle atrophy and that it participated to muscle wasting via a direct interaction with the ubiquitin ligase MAFbx[42]. Therefore, we propose that HDAC6 inhibition prevents the degradation of muscle proteins at two levels: inhibition of TGF-β signaling via Smad2/3 and inhibition of ubiquitination and subsequent degradation of MAFbx substrates. The inhibition of TGF-β signaling was also shown to be beneficial to limit cachexia and cancer progression in mouse models[119,120]. Many tumors secrete TGF-β and the finding that HDAC6 regulates TGF-β might also be relevant to explain the beneficial effects observed on cancer progression with HDAC6 inhibitors[53,54,121].

In summary, our results show that HDAC6 pharmacological inhibition significantly ameliorates several functional, biochemical, and morphological markers in skeletal muscles from dystrophin-deficient mice via three complementary mechanisms (Fig. 6): (i) microtubule stabilization that favors NMJ and DGC restoration, (ii) increase in utrophin A levels, (iii) inhibition of TGF-β signaling via Smad3 targeting which reduces muscle atrophy, fibrosis and stimulates protein synthesis. Accordingly, HDAC6 inhibition with pharmacological agents represents an attractive therapeutic target for DMD offering many advantages including ease of administration, impact on all muscles and benefits to all patients regardless of the dystrophin mutations.

## Methods

### Ethics statement, animal models, treatments, and preparation of samples

All procedures using animals were approved by the Institutional ethics committee and followed the guidelines of the National Research Council Guide for the care and use of laboratory animals, and by University of Ottawa Animal Care Committee. All procedures were in accordance with the Canadian Council of Animal Care Guidelines. For in vivo experiments, both male control C57 black 10 (C57BL10) and male C57BL/10ScSn-Dmd*mdx*/J *mdx* were used (The Jackson

laboratory Bar Harbor, USA). All animals were maintained under specific pathogen-free conditions. These mice were housed in a 12-h-light/12-h-dark cycle at a temperature controlled (23 °C ± 0.9 °C) with 50% ± 4% humidity, facility with free access to food and water. *Mdx* mice were treated for 30 consecutive days with either tubastatin A (TubA, APExBIO, #A4101; 25 mg/kg/day, intraperitoneally) solubilized in 2% DMSO or saline supplemented with 2% DMSO (vehicle-control)[46]. These mice were maintained in the Animal Care and Veterinary Service at the University of Ottawa. All animal work has complied with all relevant ethical regulations. After treatment, mice were anesthetized with isoflurane anesthesia (reference VI-1586 on device TEM SEGA, MiniTag V1 / Evaporator Tec7) and were euthanized by cervical dislocation. Then, muscles were dissected and either: (i) frozen and crushed in liquid nitrogen for protein and RNA extraction or (ii) embedded in Tissue-Tek OCT compound (VWR, Mississauga, Canada) and frozen in isopentane cooled with liquid nitrogen for cryostat sectioning[57] or (iii) were manually dissociated[80] with the following modifications: single TA fibers were fixed 30 min at room temperature in PBS-4% paraformaldehyde, permeabilized 60 min in PBS-1% Triton X-100 at 30 °C before saturation and incubation with antibodies as described below.

## Hindlimb grip strength

All injections and behavioral tests were performed in a blinded manner at the University of Ottawa Animal Behavior core facility. For TubA experiments, daily injections were continued during the behavioral testing period. To minimize interference, injections were performed in the afternoon after the completion of each test. Before each test, mice were habituated to the room for at least 30 min and tests were performed under normal light conditions. Mice were handled once a day for 3 days prior to the first test. Muscle force of each animal was measured using a Grip Strength Meter (Chatillon DFE II, Columbus Instruments) with all paws (hindlimb grip strength test). The mouse was moved closer to the meter until it had a firm grip on the probe. The mouse was pulled horizontally away from the bar at a speed of ~2.5 cm/s, until it released the probe. The value of the maximal peak force was recorded (gF). This was repeated eight times with 15 seconds intervals for each animal. In each animal at 30 day, the best score was defined as the specific maximal force. The grip strength measurements were conducted by the same investigator in order to limit variability and were performed in a random order. The investigator performing the measurements was blinded as to the treatment group of each individual mouse upon testing.

## Cell culture

C2C12 cells (ATCC, CRL-1772) were seeded on matrigel-coated (matrigel® matrix, Corning) 35 mm-diameter plates and were maintained as myoblasts in Dulbecco's modified Eagle medium (DMEM) supplemented with 10% fetal bovine serum and 1% penicillin-streptomycin (Multicell). Then cells were differentiated in differentiation medium (DMEM medium supplemented with 2% horse serum, Bio Media Canada). Cells grown in 35 mm diameter plates were treated for either Western blot or immunofluorescence. For Western blot, cells were collected by trypsinization, washed with PBS, centrifuged, and stored at −20 °C until used. For immunofluorescence, cells were fixed for 20 min in PBS-4% paraformaldehyde at room temperature, washed in PBS and stored at 4 °C until used.

## Drug, antibodies, and other reagents

C2C12 cells were treated with different drugs: TubA (at 5 μM, APExBIO, #A4101), tubacin (TBC, 5 μM, Sigma, #SML0065), rapamycin (rapa,100 nM, Sigma #CAS 53123-88-9) and SB 431542 (SB43, 10 μM, Tocris, #1614). Recombinant Human TGF-β1 (rhTGF-β1, 10 ng/mL, #100-21) was purchased from PeproTech. All primary antibodies used in this study are presented in supplementary table 1. Secondary

antibodies used for immunofluorescence studies were coupled to Alexa-Fluor 488 or Alexa-Fluor 555 (Molecular Probes); or to Cy3 or Cy5 (Jackson ImmunoResearch Laboratories). Secondary antibodies used for Western blotting were either horseradish peroxidase (HRP)-coupled anti-rabbit-IgG polyclonal antibodies (Bio-Rad) or HRP goat anti-mouse-IgG antibodies (Bio-Rad). All antibodies used in this study were validated by the manufacturers or in previously published work from the lab. To visualize NMJ for immunofluorescence studies, we used α-Bungarotoxin at 5 μg/mL conjugates with Alexa-Fluor 488 (Molecular Probes, # B13422) and DAPI (Sigma-Aldrich; # D9542) was used to stain nuclear DNA. To visualize and quantify proteins on Western blot, we used 2,2,2-Trichloroethanol[122] (TCE, Sigma, #T54801).

## Preparation of muscle and C2C12 cells homogenates

TA muscles were collected from adult mouse hindlimbs and dissected muscles were crushed on dry ice. Muscle powder resuspended in urea/thiourea buffer [7 M urea, 2 M thiourea, 65 mM chaps, 100 mM DTT, 10 U DNase I, protease inhibitors (Complete; Roche/Sigma-Aldrich)] and protein concentration was determined using CB-X Protein Assay kit (G-Bioscience, St. Louis, MO). After trypsination, C2C12 cells were solubilized in RIPA buffer [50 mM Tris−HCl, pH 8.0, 150 mM NaCl, 1% NP-40, 0.5% sodium deoxycholate, 0.1% SDS and protease inhibitors (Complete; Roche/Sigma-Aldrich)]. Protein concentration was determined using the BCA protein assay kit (Pierce/ThermoFisher Scientific) as per the manufacturer's recommendations.

## Western blot

Five to twenty μg of total proteins were separated by SDS-PAGE supplemented with 0.5% TCE and transferred onto nitrocellulose or PVDF membranes. Non-specific binding was blocked with 4% bovine serum albumin (BSA, Euromedex) diluted in 1X PBS supplemented with Tween 0.1%, and membranes were incubated with primary antibodies. After thorough washing with 0.1% Tween 1X PBS, membranes were incubated with horseradish peroxidase (HRP)-conjugated secondary antibodies (Jackson Immunoresearch Laboratories/Cederlane). After additional washes, signals were revealed using ECL substrate reagents (Bio-Rad) and acquisitions were done using a ChemiDoc™ MP Imaging Systems (Bio-Rad) or autoradiographed with X-Ray films (Fisher Scientific). Quantifications based on TCE membrane were performed with the Image Lab software (Bio-Rad) or FIJI software (ImageJ 2.0.0-rc-69/1.52n, National Institutes of Health, Bethesda, MD).

## Muscle histology, histochemistry and Masson's trichrome staining

Frozen EDL and SOL muscle samples were placed at −20 °C into the cryostat (HM 525 NX, Thermo Fisher Scientific) for at least 20 min before further processing. 10 μm thick muscle sections were transversally cut. Muscle cross-sections were stained with Hematoxylin and Eosin dyes. Sections were dehydrated using 70%, 90%, and 100% ethanol solutions and washed with toluene. The sections were mounted using Permount (Thermo Fisher Scientific) and visualized using an epifluorescent EVOS FLAuto2 inverted microscope at the University of Ottawa Cell Biology and Image Acquisition Core (RRID: SCR_021845). Percentage of central nucleation was determined by counting the total number of muscle fibers and the number of centrally nucleated muscle fibers from 6 to 8 cross-sectional views using the Northern Eclipse Software (NES, EMPIX Imaging). Masson's Trichrome staining were performed on soleus and EDL cryostat cross-sections at the University of Ottawa Louise Pelletier Histology Core Facility (RRID:SCR_021737). Sections were fixed in Bouin's solution (75 mL saturated picric acid, 25 mL 37−40% formaldehyde, 5 mL glacial acetic acid) for 1 h at 56 °C and rinsed with running water until the yellow color disappeared. Next, sections were stained in Weigert's hematoxylin (equal parts of Solution A: 1 g hematoxylin, 100 ml 95% alcohol; and Solution B: 4 ml 29% ferric chloride, 95 ml distilled water, 1 ml glacial acetic acid) for 10 min and

washed with running water for 10 min. Sections were stained in Bieb-rich's scarlet-acid fuchsin solution (90 mL 1% aqueous Biebrich scarlet, 10 ml 1% aqueous acid fuchsin, 1 mL glacial acetic acid) for 2 min and rinsed with water. Slides were incubated in phosphomolybdic-phosphotungstic acid solution (2.5 g phosphomolybdic acid, 2.5 g phosphotungstic acid, 120 mL distilled water) for 10–15 min, stained with aniline blue solution (2.5 g aniline blue, 2 ml glacial acetic acid, 100 mL distilled tilled water) for 5 min, rinsed, placed in 1% acetic acid solution for 3-5 min, dehydrated with 95% and absolute ethyl alcohol, cleared with xylene and mounted with coverslips. Sections were imaged with a Zeiss Axio Scan. Z1 equipped with a Plan-Apochromat 20x/0.8 objective. Nuclei are stained in black, cytoplasm in red and collagen in blue.

### Immunofluorescence microscopy and image acquisition

Incubations with primary antibodies in PBS-0.1% Tween 20 were per-formed either at room temperature for 60 min (C2C12 cells) or at 4 °C overnight (isolated dissociated muscle fibers) and washed. After incubation for 1–3 hours at room temperature with fluorescent sec-ondary antibodies, nuclear DNA were stained with DAPI for 10 min. Coverslips were mounted on microscope slides with FluorSave™ reagent (Calbiochem). Images were captured at RT on either a Zeiss LSM880 microscope (Carl Zeiss) with an AiryScan1 detector equipped with a 63 × 1.4-NA objective at INMG or a Zeiss Axio Imager M2 (Carl Zeiss) upright microscope equipped with either a 63 × 1.4-NA or a 10 × 0.45 NA objectives and AxioCam mRm CCD detector at the University of Ottawa Cell Biology and Image Acquisition core facility (RRID: SCR_021845). All images were processed with either the ZEN blue software, Zeiss AxioVision software (Zeiss, Oberkochen, Germany), Photoshop CS5 (Abobe Systems, San Jose, CA, USA) or FIJI software (ImageJ 2.0.0-rc-69/1.52n, National Institutes of Health, Bethesda, MD). Images were analyzed in a blinded manner by randomly renaming file names with numbers using the 'name_randomizer' macro in ImageJ[80].

### Cellular fractionation

C2C12 cells were pre-treated with TubA for 24 h and then treated for 30 min with TGF-ß1. Cells were then collected and pelleted at 300 g at 4 °C for 3 min. Pellets were resuspended with 10 volumes of ice-cold E1 buffer (20 mM NaOH pH 7.6, 5 mM Potassium Acetate, 0.5 mM MgCl2) complemented with 1 x protease inhibitor cocktail and 1 x phosphatase inhibitor cocktail and placed in 1 mL Dounce homogenizer. Cells were homogenized with 25 strokes of the loose pestle and incubated 10 min on ice before centrifugation at 2100 × g at 4 °C for 3 min. The super-natant was collected (cytoplasm extract, CE) and the pellet was resuspended in the same volume of E1 buffer supplemented of 600 mM NaCl. The suspension was incubated 30 min on ice and cen-trifuged at 20,000 × g for 20 min. The supernatant was collected (nuclear extract, NE) and the pellet was resuspended in the same volume of E1 buffer supplemented of 600 mM NaCl and Benzonase (1/1000, Millipore, 71205) and correspond to the chromatin extract (Chrm). The same volume of each fraction was loaded on Any kD Mini PROTEAN TGX gels (Biorad), transferred on nitrocellulose membrane blotting and revealed with indicated antibody.

### Chromatin immunoprecipitation

Chromatin immunoprecipitations (ChIP) of Smad2/3 protein located on the promoter sequences of atrogens (MAFbx and MuRF1) were performed using the SimpleChIP® Plus Sonication Chromatin IP Kit (Cell Signaling Technology) as followed: C2C12 cells were pre-treated or untreated with TubA for 24 h and then with TGF-ß1 for 30 min. Cells were washed in 5 mL cell culture medium (DMEM Glutamax) in the absence of Fetal Calf Serum. Cells were then incubated in DMEM Glutamax supplemented with 1% paraformaldehyde for 10 min at 25 °C. Crosslinking was stopped by adding Glycine to a 0,125 M final

concentration and incubation at 25 °C for 15 minutes. Cells were collected in PBS 1X by scraping the plates, washed in PBS 1X, and resuspended in ChIP sonication cell lysis buffer (SimpleChIP® Plus Sonication Chromatin IP Kit) containing 1X protease Inhibitor Cocktail at a concentration of 2.107 cells/mL. After 15 minutes on ice, cells were collected by centrifugation 3 minutes at 3000 g, 4 °C and resuspended again in ChIP sonication cell lysis buffer containing 1X protease Inhibitor Cocktail. After 5 minutes on ice, this last step was repeated third time for 10 minutes. Cells were sonicated in a Covaris S220 in 1 mL tube (Peak power: 140; Duty factor 10; Cycles/burst: 200) for 12 min at 4 °C. The efficiency of sonication was controlled on a 1% agarose gel to check for the presence of fragments in the range 500-1000 bp and the chromatin concentration was evaluated on a decrosslinked fraction using a nanodrop. Immunoprecipitation was performed as indicated in the ChIP CST kit protocol by incubating 2,5 µg of chromatin with 10 µL of antibody in a 500 µl final volume (adjusted with ChIP sonication lysis buffer) for 12 h at 4 °C with rotation. The ChIP antibodies used are: IgG (CST; #2729; as negative control); Histone H3 (CST; #4620, as positive control); Smad 2/3 (CST; #8685). Immunoprecipitated fragments were recovered, pur-ified and decrosslinked according to the CST kit protocol. The pre-sence of DNA were detected by QPCR (Biorad CFX Connect) using SsoADV Universal SYBR Green Supermix (Biorad) and the following primers (Eurogentec): MAFbx promoter region −205/−319 (Fbxo32-5F: 5'TTCCTTGCTACACCCTGCTT3'; Fbxo32-5R: 5'ACCTCTGCACCT CCCCTACT3'); MAFbx promoter region −424/−618 (Fbxo32-2F: 5'CT TCTTTCCCCTTCCTTTGC3'; Fbxo32-2R: 5'GGTAGGGGTGCATTCTTT GA3'); MAFbx unrelated promoter region −1232/−1320 (Fbxo32-7F:5'GGCCTGCCAGTACAGACAAT3'; Fbxo32-7R: 5'AGGTGTCTTCCT TGCTCACG3'); MuRF1 promoter region −608/−727: (Trim63-2F: 5'CA GCACAAGGGTGTTCATGT3'; Trim63-2R: 5'CTCAGTGGTAAAGGGGC TTG3'); MuRF1 unrelated promoter region −1805/−1978: (Trim63-9F: 5' GTGGATGCCAGGAACTGAAT'; Trim63-9R: 5' GGCTGTCCTGGAAC TCACTC3').

### Quantitative microtubule network lattice analysis

Using ImageJ, microtubule organization was visualized with vertical (yellow bars) and horizontal (blue bars) line scans. Using a recently developed directionality analysis tool[81] (TeDT), microtubule network lattice directionality was calculated for all mouse lines. A two-way ANOVA was used to assess the effect of microtubule intersection angle across groups. two-sided U test (Mann–Whitney) post hoc measures were used to determine the extent of differences between groups. Significance was set at $P < 0.05$.

### Quantitative analysis of compactness and fragmentation index by 'NMJ-morph' methodology

For accurate analysis, each image was captured a single en-face NMJ. NMJs that were partially oblique to the field of view were only included if the oblique portion constituted less than approximately 10% of the total area. To quantify compactness and fragmentation index, images were analyzed thanks to 'NMJ-morph' methodology[123]. The compact-ness of AChRs at the endplate was defined as follows: Compact-ness $= \left( \frac{\text{AChR area}}{\text{endplate area}} \right) \times 100$.

Fragmentation index was calculated whereby a solid plaque-like endplate has an index of (0), and highly fragmented endplate has an index that tends towards a numerical value of (1): Fragmentation index $= 1 - \left( \frac{1}{\text{number of AChR cluster}} \right)$.

The basic dimensions of the post-synaptic motor endplate were measured using standard ImageJ functions. 'NMJ-morph' is used to quantify the number of discrete AChR clusters comprising the motor endplate.

## Statistical analyses

All statistical analyses were performed using Prism 6.0 (GraphPad Software, La Jolla, USA). The nonparametric, two-sided U test (Mann–Whitney) was applied. For a multiple factorial analysis of variance, two-way ANOVA was applied. P-values under 0.05 were considered statistically significant (shown as a single asterisk in figures).

## Reporting summary

Further information on research design is available in the Nature Portfolio Reporting Summary linked to this article.

## Data availability

Source data are provided with this paper. There are no restrictions on data availability in this manuscript. The raw data in main generated in this study are provided in the Supplementary Information as a Source Data file. Supplementary figures are supplied at the end of the Supplementary Information file. Source data are provided with this paper.

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

## Acknowledgements

We thank John Lunde, Jean Luc Thomas, Amanda Tran, Laurent Coudert, Nawres Ben Marzouk and Jacques Brocard for expert technical help and fruitful discussion. We thank the PLATIM and CIQLE microscopy facilities of Lyon and the University of Ottawa Cell Biology and Image Acquisition Core (RRID: SCR_021845) for access to microscopes. Funding for this work was obtained via a grant from the Association Française contre les Myopathies (AFM telethon) to B.J.J. A.O benefited from a Postdoctoral Fellowship from the AFM during the course of this work. Additional support for this work came from AFM telethon through the strategic MyoNeurALP alliance, the Fondation pour la Recherche Médicale through an "équipe FRM" funding to L.S. and the Canadian Institutes of Health Research to B.J.J.

## Author contributions

A.O, B.J.J and L.S conceived the study, designed the project, and obtained grant funding. A.O and A.R.C performed immunofluorescence and all animal experiments. A.O and E.B performed cell fractionation experiments. A.O and I.S performed TGF-β experiments. A.O, L.M and Y.G.G. performed mTOR experiments. A.O. and P.L performed all the Western blots. A.O and V.M performed ChIP experiments. A.O performed all the culture cell experiments. A.O created Figs. 1a and 6 on Adobe Illustrator. A.O and L.S wrote the first draft manuscript. A.O, A.R.C, E.B, I.S, Y.G.G, V.M, L.M, R.M, P.L, B.J.J and L.S analyzed, interpreted the data, reviewed, finalized the manuscript, and provided comments and edits.

## Competing interests

The authors declare no competing interests.
