## [Peer Review File · Nature Communications]

Pharmacological inhibition of HDAC6 improves muscle phenotypes in dystrophin-deficient mice by downregulating TGF- β via Smad3 acetylationREVIEWER COMMENTS

Reviewer #1 (Remarks to the Author):

Duchenne muscular dystrophy (DMD) is a very common and fatal form of muscular dystrophy. There are therapeutic strategies available that have helped increase the quality of life and life expectancy of patients, however, there is as yet no cure. Research aimed to generate new therapies is very much needed and this paper is thus well placed within the field. In this paper Osseni and colleagues provide a follow up to their 2020 paper that demonstrated that HDAC6 could regulate myotubule stability and AChR clustering in muscle fibres, by applying those findings to a DMD animal model. They demonstrate that tubastatin A (an HDAC6 inhibitor) is capable of rescuing muscle strength and myofibre atrophy through improving microtubule, neuromuscular junction (NMJ), and dystrophin-associated glycoprotein complex (DGC) organisation. They further go on to show that this is mediated by both known pathways and novel mechanisms, specifically on TGF- β signalling. This research sits well in the current body of literature and should be of interest to many researchers interested not only in DMD but neuromuscular disease in general as many of these pathways are involved in other NMDs. Overall the paper is well written and this reviewer has no major concerns. There a number of areas that would benefit from more detail, slight modification, or minor rephrasing, and these are listed below.

A good introduction to DMD that clearly explains the role of the DGC in the muscle fibre membrane and how it is impacted in this condition is provided. Previous research into therapeutics is referenced and the limitations of current treatments are made clear, giving a clear rationale for this research. The pathways involved are explained in lines 60-71 and while a diagram of the proposed mechanism is given later on in the paper it would be helpful if the diagram could be expanded and used in this section to aid the reader.

Overall statistics: The authors state in the statistics section (line 752) that data distribution was assumed to be normal although that wasn't tested. Why were the data not tested for normality? The authors applied a Mann-Whitney test for many of the analysis but this is a test for data that is not normally distributed and the authors stated they assumed normality. Was this because the data did not meet their other criteria of $n > 30$? This reviewer has never seen this criteria for a t-test before – is there a reference to support its implementation? If the data is not normally distributed then presenting it as a mean would not be appropriate, median and quartiles would be more appropriate. In figures 1 c, d, e, f, g, I, l, p, plus many others, A Mann-Whitney U test has been used to compare data. However this test is to compare 2 columns of data – this point would be raised in the Prism software that the authors are using. If they performed pairwise comparisons on each group, then it is important that they correct for multiple comparisons and state this has been performed in the figure legend.

The graphs show mean \pm SEM – this reviewer would suggest changing this to SD as this is more appropriate (Drummond & Tom, 2011. J of Phys)

Extended data Fig 1.a – Please could the authors give more details in the figure legend as to how this WB was performed? The blots for ac-H3 and H3 look very similar with just a slight change in exposure. Is this the same membrane striped and re-probed? Is there another reason why the bands look so similar?

Fig 1f Looks to be measuring force gained over time by each animal. Would a statistical test that takes this into account not be more appropriate e.g. a paired t-test for 2 columns of data, or a matched/repeated measures anova for more than one column?

Fig 1 – Define TCE at first mention & in text in methods.

Line 112 – define TA at first mention

The fact that grip strength overall was increased Fig 1e, but specific maximal force was not, is interesting. Do the authors think there could be a differential effect on strength vs muscle fatigue? Was there a drop in grip strength measured over the 5 pulls?

Line 152 – While the TubA treatment has clearly improved β -DG levels at the plasma membrane in mdx mice, this reviewer would caution against the use of the word 'restore' here, as that suggests that the levels are back to WT, which they do not appear to be.

Line 168 – Would suggest having the figures cited in the order they appear in the actual figure.
Line 176 – The authors refer to an overall increase in fibre size. However this is not what was observed. The data shows a reduction in very small fibres and more of a normalisation of fibre size. For the soleus the median values of mdx-TubA fibres was less than the untreated, and there was an accumulation of fibres in the very large CSA bins for the untreated animals. I would suggest that this sentence to be rephrased to suggest a reduction in atrophy rather than an increase in fibre size.
Figure 2 – Some of the graphs show WT data and others don't, is there a reason for this?
Figure 2b, e – The boxes show the median values and there are significance stars next to the mdx-TubA group – what was this in comparison to?
Line 194 – Figure 2 m & P – This should be Fig 2 n & p.
Figure 3 b – It is unclear which groups the significance star is in relation to and it is not stated in the legend.
Line 211-220– While the lattice has been improved considerably, it does not appear completely restored, would caution about the use of the word 'restored' here.
Line 220 – Do the authors mean 'Described'
Figure 3d – in the box the * next to the treated animals is compared to which other group?
The authors used NMJ-Morph to perform the analysis and there is a strong treatment effect observed. NMJ-Morph allows for the measurement of around 20 variables, many of which could be measured with just the α -Bun staining (AChR area/perimeter, Endplate perimeter/area). Could the authors add in more detail as to why they chose these variables to be included? Or even include all the data in the extended data file.
Line 241 – 'changes' > changed
Figure 4 l – legend text 'were been'
Line 650 – would this not be all paws grip strength rather than hindlimb? If so then change in results graphs.
Line 655 'animals' > animal
Line 679 – the words 'with either' dont make sense in this sentence.
Line 718 – Do the authors mean 'nuclear DNA were stained'?

Reviewer #2 (Remarks to the Author):

The manuscript of Osseni et al. describes the positive effect of a 30-day treatment of the mdx mouse model for Duchenne muscular dystrophy with a selective HDAC6 inhibitor (tubastatin A). This results in a significant amelioration of the phenotype of these mice (restoration of muscle strength; protection against muscle atrophy, reduction of the number of centronucleated muscle fibers; reduction of the proportion of small fibers; lowering of fibrosis). Moreover, tubastatin A treatment also restored the microtubule lattice. This positive effect is correlated with an upregulation of utrophin A and the restoration of the DGC complex. Tubastatin A also affects the mTOR pathway and an acetylation of Smad2/3 is observed. Overall it is concluded that a regulation of the TGF-beta signaling by increasing the Smad2/3 acetylation is responsible for the beneficial effect of HDAC6 inhibition in the mdx mouse model.

Major remarks

- While the therapeutic effect of HDAC6 inhibition on the dystrophin-deficient mice is impressive and well characterized, the exact mechanism(s) underlying this beneficial effect is less clear. There is a correlation with the upregulation of utrophin A (although this is not the nicest blot) and the restoration of the DGC complex. However, the mechanistic (and causal) link with the mTOR pathway and the acetylation of Smad2/Smad3 is less convincing. In addition, more efforts should be done to explain the experiments documenting these potential link(s) in a more logical and systematic way.
- There seems to be a potential contradiction between the effects of HDAC6 inhibition observed on AChR clustering as reported in the JCB paper of the same research group (...pharmacological inhibition of HDAC6 protects against MT disorganization and reduces the size of acetylcholine receptor (AChR) clusters) and the positive effect on the NMJs reported in the current manuscript (line 236: an increase

of the compactness and recruitment of AchR patches). This apparent contradiction should be explained in more detail.

Minor remarks

- The sentence in the abstract starting with 'Conversely, to inhibitors...' is difficult to understand.
- The introduction is rather long and not very focused. Especially the paragraph/information on the pan-deacetylase inhibitors could be shortened/removed as this could potentially confuse the readers.
- All citations should be carefully checked. One example is ref 44. While it is indicated in the introduction that HDAC6 inhibition is beneficial for ALS, this paper only contains a crossbreeding experiment with an ALS mouse model and no pharmacological inhibition of HDAC6. Having said that, there are other (=better) examples of beneficial effects of HDAC6 inhibition in the context of ALS.
- It is not clear why the result section is not subdivided.
- When the Y-axis of a figure doesn't start at 0, it shouldn't be connected to the X-axis (as this could be misleading). Leaving some space in between is sufficient.
- The quality of the Smad2/Smad3 Western blot in panel f of Figure 5 is low as the separation of the bands is difficult to see.
- The manuscript contains several typos which should all be corrected. There are many examples. One annoying one is the description of the mdx mouse model (line 105: dystrophic-deficient mice; line 368: dystrophic dystrophic-deficient muscles).
- Not clear why 'via' in alleviated is in italic.

Reviewer #3 (Remarks to the Author):

In this paper "Pharmacological inhibition of HDAC6 downregulates TGF- β via Smad2/3 acetylation and improves dystrophin deficient muscles" Osseni et al. tested HDAC6-specific inhibitor, TubastatinA (TubA), as a possible pharmacological approach to ameliorate dystrophic muscles. To this end, they used the well-established model of Mdx mice treated for 30 days with TubA and analyzed different histological and functional parameters.

The results convincingly demonstrate that TubA treatment ameliorates dystrophic muscles both at the morphological and functional level. They provide insights into a number of complementary mechanisms that might explain these TubA-mediated beneficial effects, which however I found rather disconnected and confound the flow of the article.

For instance, is the upregulation of Utrophin and b-DG mediated by the induction of mTOR? Does this depend on the general increase in protein synthesis? The authors could easily test this in the setting shown in Fig. 1n-o upon downregulation/inhibition of mTOR.

Also, the mechanism behind mTOR activation upon HDAC6 inhibition is rather obscure. The authors tried to link TubA-induced mTOR signalling to TGF β pathway inhibition. However, they do not test this possible functional crosstalk/antagonism experimentally in the context of HDAC6 inhibition. They rely on the observation that TubA induces Smad2/3 acetylation, which they suggest counteracts Smad2/3 phosphorylation and therefore prevents expression of target genes (i.e. atrogenes) while also promoting mTOR signaling. However, it is not clear to me how this is achieved. Does the acetylation of Smad2/3 inhibit their translocation to the nucleus? Does this prevent their binding to the chromatin? The authors suggest this in the discussion, but I think this aspect should be experimentally verified. For example, a biochemical fractionation (cytosolic, nuclear and chromatin) of cells +/- TubA could help answering this question. Moreover, is acetylation, or the global level, of Smad7 (the inhibitor of TGF β pathway) affected by TubA?

That said, I think the paper is very well-written, the data are well presented and statistically sound.

The results are of significance in the field, as they expand the current knowledge and possible translational potential of pharmacological approaches aimed at inhibiting specific HDACs.

However, my enthusiasm was hampered by the lack of a clear mechanism, so I suggest the authors to make a bigger effort to try linking, experimentally, at least some of their results. I would also be curious to know whether this compound recovers the dystrophic phenotype if administered at later

stages of the disease, when the muscles are already degenerated. It is known that pan-HDACi are not effective at this stage, and this could represent a major advantage over existing drugs in trials (i.e. givinostat)

Additional specific points:

- In figure 5f-i, they should also show the levels of ac-Smad2/3 and p-Smad2/3 in wt muscles, to verify the extent of acetylated-Smad in normal muscles.

The WB shown in Fig. 5f and h display higher amounts of total Smad2/3, is this an actual induction? WB for a normalizer should be also shown.

- The authors show histo-morphological analyses only on EDL and Soleus muscles. However, I think it is important to also assess the impact of TubA on diaphragm, which is the most affected muscle in Mdx mice.

- Fig2n/p: although the authors assessed the global levels of fibrotic proteins (collagens and CTGF) by WB. I think that also showing the extent of fibrotic infiltration histologically (i.e. Sirius red; Masson's trichrome etc) would be useful to assess the extent of interstitial fibrotic deposit.

REVIEWER COMMENTS

Reviewer #1 (Remarks to the Author):

Duchenne muscular dystrophy (DMD) is a very common and fatal form of muscular dystrophy. There are therapeutic strategies available that have helped increase the quality of life and life expectancy of patients, however, there is as yet no cure. Research aimed to generate new therapies is very much needed and this paper is thus well placed within the field. In this paper Osseni and colleagues provide a follow up to their 2020 paper that demonstrated that HDAC6 could regulate myotubule stability and AChR clustering in muscle fibres, by applying those findings to a DMD animal model. They demonstrate that tubastatin A (an HDAC6 inhibitor) is capable of rescuing muscle strength and myofibre atrophy through improving microtubule, neuromuscular junction (NMJ), and dystrophin-associated glycoprotein complex (DGC) organisation. They further go on to show that this is mediated by both known pathways and novel mechanisms, specifically on TGF- β signalling. This research sits well in the current body of literature and should be of interest to many researchers interested not only in DMD but neuromuscular disease in general as many of these pathways are involved in other NMDs.

Overall the paper is well written and this reviewer has no major concerns. There are a number of areas that would benefit from more detail, slight modification, or minor rephrasing, and these are listed below. A good introduction to DMD that clearly explains the role of the DGC in the muscle fibre membrane and how it is impacted in this condition is provided. Previous research into therapeutics is referenced and the limitations of current treatments are made clear, giving a clear rationale for this research.

We thank the reviewer for insightful comments and for carefully reading our manuscript. We were pleased to read that this reviewer found that our work sits well in the current literature, that it should be of interest to many researchers, and that the paper is well written with no major concerns.

The pathways involved are explained in lines 60-71 and while a diagram of the proposed mechanism is given later on in the paper it would be helpful if the diagram could be expanded and used in this section to aid the reader.

Thank you for this comment. We have modified this section to clarify our message to help readers line 61-72.

Overall statistics: The authors state in the statistics section (line 752) that data distribution was assumed to be normal although that wasn't tested. Why were the data not tested for normality? The authors applied a Mann-Whitney test for many of the analysis but this is a test for data that is not normally distributed and the authors stated they assumed normality. Was this because the data did not meet their other criteria of $n > 30$? This reviewer has never seen this criteria for a t-test before – is there a reference to support its implementation? If the data is not normally distributed then presenting it as a mean would not be appropriate, median and quartiles would be more appropriate.

In figures 1 c, d, e, f, g, l, l, p, plus many others, A Mann-Whitney U test has been used to compare data. However this test is to compare 2 columns of data – this point would be raised in the Prism software that the authors are using. If they performed pairwise comparisons on each group, then it is important that they correct for multiple comparisons and state this has been performed in the figure legend. The graphs show mean \pm SEM – this reviewer would suggest changing this to SD as this is more appropriate (Drummond & Tom, 2011. J of Phys)

We have complied with the suggestion. In addition, we have used the Prism software, boxes and whiskers to illustrate all quantifications in this article. For enhanced consistency, we used a nonparametric Mann-Whitney test, which does not assume Gaussian distributions. For multiple comparisons, we have used Two-Way ANOVAs.

Extended data Fig 1.a – Please could the authors give more details in the figure legend as to how this WB was performed? The blots for ac-H3 and H3 look very similar with just a slight change in exposure. Is this the same membrane striped and re-probed? Is there another reason why the bands look so similar?

Indeed, for this control Western blot in Extended data Figure 1a, the same membrane was striped and re-probed. This is now mentioned in the figure legend.

Fig 1f Looks to be measuring force gained over time by each animal. Would a statistical test that takes this into account not be more appropriate e.g. a paired t-test for 2 columns of data, or a matched/repeated measures anova for more than one column?

Indeed, it is more appropriate to use a paired *t*-test for 2 columns and we changed the text according to this comment (see figure legend 1).

Fig 1 – Define TCE at first mention & in text in methods.

As requested, TCE is now defined at first mention in Figure 1.

Line 112 – define TA at first mention

As requested, TA is now defined at first mention lines 111-112.

The fact that grip strength overall was increased Fig 1e, but specific maximal force was not, is interesting. Do the authors think there could be a differential effect on strength vs muscle fatigue? Was there a drop in grip strength measured over the 5 pulls?

This is a very good point. We had already performed experiments to evaluate the effect of grip strength over 8 pulls after 30 days treatment. We have added a new figure in Extended data Fig. 1e to show this. We observe in *mdx*-vehicle a drop in grip strength indicating fatigue, whereas this drop was not observed in *mdx* treated with TubA. This is now indicated in the text (lines 131 to 137).

Line 152 – While the TubA treatment has clearly improved β -DG levels at the plasma membrane in mdx mice, this reviewer would caution against the use of the word ‘restore’ here, as that suggests that the levels are back to WT, which they do not appear to be.

The sentence was reformulated accordingly (line 158).

Line 168 – Would suggest having the figures cited in the order they appear in the actual figure.

We have changed figure 2 accordingly (figure 2 and lines 175, 177 and 180).

Line 176 – The authors refer to an overall increase in fibre size. However this is not what was observed. The data shows a reduction in very small fibres and more of a normalisation of fibre size. For the soleus the median values of mdx-TubA fibres was less than the untreated, and there was an accumulation of fibres in the very large CSA bins for the untreated animals. I would suggest that this sentence to be rephrased to suggest a reduction in atrophy rather than an increase in fibre size.

Indeed, compared with wild type muscles, *mdx* muscles present abnormal fiber size distribution with an increase of fiber size heterogeneity. We have modified the section according to the reviewer’s comment (lines 175 and 182).

Figure 2 – Some of the graphs show WT data and others don’t, is there a reason for this?

In some case and especially in CSA pictures, we feel it is important to show WT to better highlight the impact of HDAC6 inhibition on normal histology. So, we have included the WT data where most relevant to illustrate the beneficial effect of TubA.

Figure 2b, e – The boxes show the median values and there are significance stars next to the mdx-TubA group – what was this in comparison to?

To clarify this, lines were added on Figures 2c, d (formerly Figures 2b, e) to better define comparison of median values between *mdx-veh* mice and *mdx-TubA* mice. This is now mentioned in the figure legend as well.

Line 194 – Figure 2 m & P – This should be Fig 2 n & p.

Thank you for your comment. We corrected this oversight in Figure 2.

Figure 3 b – It is unclear which groups the significance star is in relation to and it is not stated in the legend.

This point was clarified in figure 3b.

Line 211-220– While the lattice has been improved considerably, it does not appear completely restored, would caution about the use of the word ‘restored’ here.

The sentence was rephrased accordingly (line 231)

Line 220 – Do the authors mean ‘Described’

Thanks for your comment. We removed this sentence.

Figure 3d – in the box the * next to the treated animals is compared to which other group?

This is now specified in the legend of figure 3.

The authors used NMJ-Morph to perform the analysis and there is a strong treatment effect observed. NMJ-Morph allows for the measurement of around 20 variables, many of which could be measured with just the α -Bun staining (AChR area/perimeter, Endplate perimeter/area). Could the authors add in more detail as to why they chose these variables to be included? Or even include all the data in the extended data file.

Indeed, based and adapted from the paper of Jones et al., (Open Biol. 2016 Dec;6(12):160240. doi: 10.1098/rsob.160240), more than 20 variables could be measured. However, in *mdx* mice, only the shape of the NMJ on the post-synaptic elements is very affected. So, we focused on classical features of the post-synaptic domain. As requested, we nonetheless added all the data concerning AChR area and endplate perimeter in the extended data figure 3f and 3g.

Line 241 – ‘changes’ > changed

This typo was corrected (line 254).

Figure 4 I – legend text ‘were been’

This oversight was corrected in the legend of Figure 4I.

Line 650 – would this not be all paws grip strength rather than hindlimb? If so then change in results graphs.

The graphs were changed (Figure 1).

Line 655 ‘animals’ > animal

This typo was corrected (line 765).

Line 679 – the words ‘with either’ dont make sense in this sentence.

The word ‘either’ was removed (line 791).

Line 718 – Do the authors mean ‘nuclear DNA were stained’?

The sentence was rephrased (line 846).

Reviewer #2 (Remarks to the Author):

The manuscript of Osseni et al. describes the positive effect of a 30-day treatment of the mdx mouse model for Duchenne muscular dystrophy with a selective HDAC6 inhibitor (tubastatin A). This results in a significant amelioration of the phenotype of these mice (restoration of muscle strength; protection against muscle atrophy, reduction of the number of centronucleated muscle fibers; reduction of the proportion of small fibers; lowering of fibrosis). Moreover, tubastatin A treatment also restored the microtubule lattice. This positive effect is correlated with an upregulation of utrophin A and the restoration of the DGC complex. Tubastatin A also affects the mTOR pathway and an acetylation of Smad2/3 is observed. Overall it is concluded that a regulation of the TGF-beta signaling by increasing the Smad2/3 acetylation is responsible for the beneficial effect of HDAC6 inhibition in the mdx mouse model.

We thank this reviewer for carefully reading through our paper and for providing constructive comments which have clarified key issues.

Major remarks

- While the therapeutic effect of HDAC6 inhibition on the dystrophin-deficient mice is impressive and well characterized, the exact mechanism(s) underlying this beneficial effect is less clear. There is a correlation with the upregulation of utrophin A (although this is not the nicest blot) and the restoration of the DGC complex. However, the mechanistic (and causal) link with the mTOR pathway and the acetylation of Smad2/Smad3 is less convincing. In addition, more efforts should be done to explain the experiments documenting these potential link(s) in a more logical and systematic way.

Thank you for your comment. We initially did not investigate this aspect because the mechanisms linking mTOR and TGFb/Smad pathways are still poorly understood. Nevertheless, following your comment, we put more efforts to address this and provide an explanation for the effect of Smad inhibition on mTOR signaling. In 2013, Goodman et al. showed that in parallel with MAFbx translational activation, Smad3 activates translation of PTEN mRNA, thereby inhibiting Akt/mTOR signaling and protein synthesis (Goodman et al, Mol Endocrinol. 2013 Nov;27(11):1946-57. doi: 10.1210/me.2013-1194). We therefore examined as part of the current work whether downregulation of PTEN expression occurred following Smad 3 inhibition by TubA. We now show that compared to controls, PTEN protein levels are decreased following TubA treatment, and Akt protein levels and phosphorylation are increased. These new results are included in Extended Data Fig. 5i-s and in the text (lines 325 to 334 and 394 to 399)

- There seems to be a potential contradiction between the effects of HDAC6 inhibition observed on AchR clustering as reported in the JCB paper of the same research group (...pharmacological inhibition of HDAC6 protects against MT disorganization and reduces the size of acetylcholine receptor (AChR) clusters) and the positive effect on the NMJs reported in the current manuscript (line 236: an increase of the compactness and recruitment of AchR patches). This apparent contradiction should be explained in more detail.

We thank the reviewer for raising this important point. Indeed, in 2020 we reported that "pharmacological inhibition of HDAC6 protects against MT disorganization and reduces the size of acetylcholine receptor (AChR) clusters". We do not believe that our new results are contradictory. Our previous and new results both show that inhibiting HDAC6 increases the compactness and reduces the perimeter of the NMJ. In the context of DMD, NMJ are abnormally large because of fragmentation and our results indicate that stabilizing the microtubule network limits fragmentation, which is consistent with our previous results that show that increasing microtubule acetylation limits the spreading of the NMJ. This point is now discussed in the text (lines 238 to 249 and 368 to 372)

Minor remarks

- The sentence in the abstract starting with 'Conversely, to inhibitors...' is difficult to understand.

Relevant sentences in the abstract have been reformulated.

- The introduction is rather long and not very focused. Especially the paragraph/information on the pan-deacetylase inhibitors could be shortened/removed as this could potentially confuse the readers.

Thank you for this comment. As requested, and to clarify the information, we have shortened this paragraph (see line 61).

- All citations should be carefully checked. One example is ref 44. While it is indicated in the introduction that HDAC6 inhibition is beneficial for ALS, this paper only contains a crossbreeding experiment with an ALS mouse model and no pharmacological inhibition of HDAC6. Having said that, there are other examples of beneficial effects of HDAC6 inhibition in the context of ALS.

Thank you for bringing this up to our attention. All citations were carefully checked and rephrasing of the sentence (lines 81 to 83) was done. A new citation was also added.

- It is not clear why the result section is not subdivided.

As suggested, the result section is now subdivided

- When the Y-axis of a figure doesn't start at 0, it shouldn't be connected to the X-axis (as this could be misleading). Leaving some space in between is sufficient.

All graphs with this were adjusted according to the reviewer's comment (Fig 1e, f, g and Fig 2e, f).

- The quality of the Smad2/Smad3 Western blot in panel f of Figure 5 is low as the separation of the bands is difficult to see.

We have added TCE as requested by Reviewer 3 and changed the relevant figures so that now, the separation of bands is clearer as it could be seen in figure 5 and extended data figure 5.

- The manuscript contains several typos which should all be corrected. There are many examples. One annoying one is the description of the mdx mouse model (line 105: dystrophic-deficient mice; line 368: dystrophic dystrophic-deficient muscles).

Thank you for pointing this out. We apologize for the many typos and have gone over the manuscript carefully to make necessary corrections. In particular, rephrasing of this sentence was done (line 451 in the new version).

- Not clear why 'via' in alleviated is in italic.

This section was removed, and thanks to your previous comment, we have shortened this paragraph as requested (line 61).

Reviewer #3 (Remarks to the Author):

In this paper “Pharmacological inhibition of HDAC6 downregulates TGF-B via Smad2/3 acetylation and improves dystrophin deficient muscles” Osseni et al. tested HDAC6-specific inhibitor, TubastatinA (TubA), as a possible pharmacological approach to ameliorate dystrophic muscles. To this end, they used the well-established model of Mdx mice treated for 30 days with TubA and analyzed different histological and functional parameters. The results convincingly demonstrate that TubA treatment ameliorates dystrophic muscles both at the morphological and functional level. They provide insights into a number of complementary mechanisms that might explain these TubA-mediated beneficial effects, which however I found rather disconnected and confound the flow of the article.

We thank the reviewer for carefully providing important comments to enhance the clarity of our work. We were delighted to read that this reviewer found our data to be convincing and that we provide insights into many mechanisms.

For instance, is the upregulation of Utrophin and b-DG mediated by the induction of mTOR?

We also agree with the last statement. However, given the established role of Utrophin in mdx muscles as a compensatory protein, we felt that it was important to include these data in the current paper. To more specifically address the point on mTOR, new experiments were performed. In this revised version, we now show that mTOR inhibition by rapamycin prevents the increase in Utrophin levels (lines 271 to 276). These additional findings were added in Extended data Fig. 4e, 4f.

Does this depend on the general increase in protein synthesis? The authors could easily test this in the setting shown in Fig. 1n-o upon downregulation/inhibition of mTOR.

Thank you for your comment. As part of the revisions for our paper, we have investigated how HDAC6 inhibition could activate mTOR signaling (see also major remark #1 from Reviewer 2). We found that compared to controls, PTEN protein levels are decreased upon TubA exposure and Akt protein levels and phosphorylation are increased. This now provides an explanation on how HDAC6 inhibition increases mTOR signaling and protein translation. These new results are included in Extended Data Fig. 5i-s and in lines 325 to 334.

In addition, we have investigated the effect of the mTORC1 inhibitor Rapamycin and found that TubA cannot bypass a direct inhibition of the mTORC1 complex. This is consistent with the fact that we now also show that HDAC6 inhibition acts on PTEN and Akt, upstream of mTOR. New panels have been added in Extended data Fig. 4b, 4c, 4d, to show that in the presence of rapamycin, tubastatin A does not prevent the decrease of S6 phosphorylation (lines 263 to 270)

We further evaluated the effects of rapamycin on Utrophin A expression by Western blot using C2C12 whole cell extracts. As expected, TubA alone increased Utrophin A levels. However, in the presence of rapamycin, Utrophin A levels were not increased by TubA (extended data Fig. 4e and 4f). Altogether, these new data indicate that the

increase in utrophin A expression caused by TubA exposure is mediated via mTOR signaling. This is now included in the result section (lines 271 to 276).

Also, the mechanism behind mTOR activation upon HDAC6 inhibition is rather obscure. The authors tried to link TubA-induced mTOR signaling to TGFb pathway inhibition. However, they do not test this possible functional crosstalk/antagonism experimentally in the context of HDAC6 inhibition. They rely on the observation that TubA induces Smad2/3 acetylation, which they suggest counteracts Smad2/3 phosphorylation and therefore prevents expression of target genes (i.e. atrogenes) while also promoting mTOR signaling. However, it is not clear to me how this is achieved. Does the acetylation of Smad2/3 inhibit their translocation to the nucleus? Does this prevent their binding to the chromatin?

Thank you for this important comment. As detailed in the above sections, we have now included data showing how HDAC6 inhibition activates the TOR pathways via PTEN inhibition.

With regards to the effect of Smad3 acetylation, we show that it prevents Smad2/3 phosphorylation which is a prerequisite for Smad nuclear translocation. In our view, an unbiased piece of evidence of inhibition is the ability of Smad 2/3 to activate their target. This can be investigated by determining the recruitment of Smad2/3 onto the promoters of target genes. Accordingly, we have specifically investigated this using chromatin immunoprecipitation (ChIP) to visualize Smad2/3 recruitment on the promoters of MAFbx and MURF1. ChIP experiments were performed on C2C12 muscle cells treated with TGFβ1 in the presence or absence of TubA. QPCR amplification of the promoters of MAFbx and MURF1 on immunoprecipitated DNA showed that TubA indeed reduces Smad2/3 recruitment onto chromatin. This important new result has been added to the results section (lines 304 to 313) and in Figures 5i and 5j.

The authors suggest this in the discussion, but I think this aspect should be experimentally verified. For example, a biochemical fractionation (cytosolic, nuclear and chromatin) of cells +/- TubA could help answering this question.

Thank you for your comment. Cell fractionation assays were performed in the presence or absence of TubA to separate cytosolic fraction (CE) and a nuclear fraction with the latter being further sub-divided into two parts: a nuclear extract (NE) and a chromatin extract (Chrm). As expected, we observed that the cytoplasmic/nuclear ratio of phosphor-Smad was inverted in the TubA-treated cells compared to controls. A new panel was added in Figure 5f and see lines 298 to 303 to highlight this.

Moreover, is acetylation, or the global level, of Smad7 (the inhibitor of TGFb pathway) affected by TubA?

As suggested, Smad7 levels were evaluated. We also determined the level of the common Smad mediator Smad4. TubA induced no significant change in the levels of Smad4 and Smad7. These new results are now included in Extended data Fig. 5e-h and lines 322-324 in the text.

That said, I think the paper is very well-written, the data are well presented and statistically sound. The results are of significance in the field, as they expand the current knowledge and possible translational potential of pharmacological approaches aimed at inhibiting specific HDACs. However, my enthusiasm was hampered by the lack of a clear mechanism, so I suggest the authors to make a bigger effort to try linking, experimentally, at least some of their results. I would also be curious to know whether this compound recovers the dystrophic phenotype if administered at later stages of the disease, when the muscles are already degenerated. It is known that pan-HDACi are not effective at this stage, and this could represent a major advantage over existing drugs in trials (i.e. givinostat).

We thank the reviewer for the many very insightful comments. We are pleased that this reviewer feels that our paper is well written, that the data are well presented and that the results are of significance. As detailed above and in the revised manuscript, we have made considerable efforts to systematically address all the concerns raised by the reviewers, especially to try to better link our findings with new mechanistic data.

We agree that it will be important in future work to evaluate if HDAC6 inhibitors are efficient at late stages of the disease. The results are not granted since at later stages of DMD, especially when fibrosis is set in. We believe that selective inhibitors of HDAC6 represent to this day a good alternative to pan-HDAC inhibitors. Indeed, tubastatin A increases CSA of muscle fibers, reduces fibrosis (our results and Sin Young Choi et al., *Vascul Pharmacol.* 2015 Sep;72:130-40. doi: 10.1016/j.vph.2015.04.006) and reduces inflammation (Santosh Vishwakarma, *Int Immunopharmacol.* 2013 May;16(1):72-8. doi: 10.1016/j.intimp.2013.03.016). In addition, mice lacking HDAC6 are viable and do not develop any phenotype in normal conditions (Yu Zhang, *Mol Cell Biol.* 2008 Mar;28(5):1688-701. doi: 10.1128/MCB.01154-06.), supporting the idea that HDAC6 is an interesting long-term pharmacological target.

Additional specific points:

- In figure 5f-i, they should also show the levels of ac-Smad2/3 and p-Smad2/3 in wt muscles, to verify the extent of acetylated-Smad in normal muscles.

As requested, we have added the levels of ac-Smad2/3 and p-Smad2/3 in WT muscles in figure 5k, n. We did not observe a significant difference in Smad acetylation between WT and *mdx* mice.

The WB shown in Fig. 5f and h display higher amounts of total Smad2/3, is this an actual induction? WB for a normalizer should be also shown.

Thank you for your comment. As requested, we have added a normalizer (Total protein amount, TCE) in figures 5k, n. TubA treatment indeed seems to increase the total levels of Smad 2/3, but our measures did not reach a satisfactory statistical significance ($P = 0.0571$; Fig. 5n). This is now mentioned in the text (lines 320 to 322).

- The authors show histo-morphological analyses only on EDL and Soleus muscles. However, I think it is important to also assess the impact of TubA on diaphragm, which is the most affected muscle in Mdx mice.

Indeed, the diaphragm is the most affected muscle in *mdx* mice. For histo-morphological analyses, we focused this study with 2 antagonist muscles widely used in *mdx* mice: the slow oxidative SOLEUS muscle and the fast glycolytic extensor digitorum longus muscle (EDL). We did not harvest the diaphragm muscle for histo-morphological analyses. Ethically, it is difficult for us to justify sacrificing more mice to harvest diaphragm muscles because positive effects in EDL and SOL muscles were already observed and quantified. Indeed, our institutional ethical committee will not let us use another mouse cohort to add the diaphragm to our findings.

- Fig2n/p: although the authors assessed the global levels of fibrotic proteins (collagens and CTGF) by WB. I think that also showing the extent of fibrotic infiltration histologically (i.e. Sirius red; Masson's trichrome etc) would be useful to assess the extent of interstitial fibrotic deposit.

Thank you for this comment. As requested, to evaluate the levels of collagen infiltration, Masson's trichrome staining was performed. New figures were added in Figure 2r, 2s and in Extended data Fig. 2a, 2b and lines 204-209 in the text.

REVIEWERS' COMMENTS

Reviewer #1 (Remarks to the Author):

The authors have responded to all comments

Reviewer #2 (Remarks to the Author):

The authors answered all my questions and adapted the manuscript accordingly. I have no further remarks.

Reviewer #3 (Remarks to the Author):

The authors satisfactorily addressed my concerns.